# Chemical characterization of fine particulate matter emitted by peat fires in Central Kalimantan, Indonesia, during the 2015 El Niño

Thilina Jayarathne[1], Chelsea E. Stockwell[2], Ashley A. Gilbert[1], Kaitlyn Daugherty[1], Mark A. Cochrane[3], Kevin C. Ryan[4], Erianto I. Putra[3,5], Bambang H. Saharjo[5], Ati D. Nurhayati[5], Israr Albar[5,a], Robert J. Yokelson[6] and Elizabeth A. Stone[1,7]

[1]Department of Chemistry, University of Iowa, Iowa City, IA 52242, USA
[2]Chemical Science Division, NOAA Earth System Research Laboratory, Boulder, 80305, USA
[3]South Dakota State University, Geospatial Science Center of Excellence, Brookings, 57006, USA
[4]FireTree Wildland Fire Sciences, L.L.C., Missoula, 59801, USA
[5]Bogor Agricultural University, Faculty of Forestry, Bogor, 16680, ID
[6]University of Montana, Department of Chemistry, Missoula, 59812, USA
[7]Chemical and Biochemical Engineering, University of Iowa, Iowa City, IA 52242, USA
[a]Now at: Climate Change Division, Ministry of Environmental and Forestry, Jakarta 10270, ID

*Correspondence to*: Elizabeth A. Stone (betsy-stone@uiowa.edu)

**Abstract.** Fine particulate matter ($PM_{2.5}$) was collected *in situ* from peat smoke during the 2015 El Niño peat fire episode in Central Kalimantan, Indonesia. Twenty-one PM samples were collected from 18 peat fire plumes that were primarily smoldering with modified combustion efficiency (MCE) values of 0.725-0.833. PM emissions were determined and chemically characterized for elemental carbon (EC), organic carbon (OC), water-soluble OC, water-soluble ions, metals, and organic species. Fuel-based $PM_{2.5}$ mass emission factors (EF) ranged from 6.0 - 29.6 g kg$^{-1}$ with an average of 17.3±6.0 g kg$^{-1}$. EC was detected only in 15 plumes and comprised ~1% of PM mass. Together, OC (72 %), EC (1 %), water-soluble ions (1 %) and metal oxides (0.1 %) comprised 74±11 % of gravimetrically-measured PM mass. Assuming that the remaining mass is due to elements that form organic matter (OM; i.e. elements O, H, N) an OM to OC conversion factor of 1.26 was estimated by linear regression. Overall, chemical speciation revealed the following characteristics of peat burning emissions: high OC mass fractions (72 %), primarily water-insoluble OC (84±11 %C), low EC mass fractions (1 %), vanillic to syringic acid ratios of 1.9, and relatively high n-alkane contributions to OC (6.2 %C) with a carbon preference index of 1.2-1.6. Comparison to laboratory studies of peat combustion revealed similarities in the relative composition of PM, but greater differences in the absolute EF values. The EF developed herein, combined with estimates of the mass of peat burned, are used to estimate that 3.2 - 11 Tg of $PM_{2.5}$ was emitted to atmosphere during the 2015 El Niño peatland fire event in Indonesia. Combined with gas-phase measurements of $CO_2$, CO, $CH_4$ and VOC from Stockwell et al. (2016), it is determined that OC and EC account for 2.1 % and 0.04 % of total carbon emissions, respectively. These *in situ* EFs can be used to improve the accuracy of the representation of Indonesian peat burning in emission inventories and receptor-based models.

## 1 Introduction

In recent decades, peatland fires in Southeast Asia, especially the Indonesian provinces of Sumatra, Kalimantan, and Papua as well as Malaysian Borneo have become more frequent in occurrence (Page et al., 2009; Langner and Siegert, 2009; Van der Werf et al., 2010). The 2015 El Niño-driven peatland fire episode that occurred September – October 2015 was more extensive than in normal years and reported as the next-strongest peatland fire episode since 1997-98 (Parker et al., 2016; Koplitz et al., 2016; Huijnen et al., 2016). The 2015 fires burned ~1 million hectares of tropical forests and peatlands in Indonesia, releasing ~0.2 Pg C of carbon to the atmosphere (Huijnen et al., 2016). However, these values are well below the 1997-98 records of ~2 million hectares of burned peatland area that released ~1.7 Pg C of carbon to the atmosphere, due to early monsoons and more effective fire control strategies in 2015 (Page et al., 2002; Chisholm et al., 2016; Huijnen et al., 2016; Tacconi, 2003). The direct effects of 2015 peatland fire smoke affected neighboring Singapore, Malaysia, Thailand and Philippines with an estimated economic loss greater than 16 billion USD to their GDPs due to declines in productions and services during the event, and long-term impacts to human health and the environment (Glover and Jessup, 2006; Chisholm et al., 2016; WorldBank, 2016). Negative health effects due to inhalation of peat smoke were widely reported during this catastrophe (Koplitz et al., 2016). In Palangka Raya, the capital of Central Kalimantan, $PM_{10}$ levels reached up to 3741µg $m^{-3}$, nearly two orders of magnitude higher than the World Health Organization (WHO) guideline for 24 hour $PM_{10}$ exposure (Stockwell et al., 2016; WHO, 2005). It has estimated that more than 40 million people suffered from continuous exposure to peat smoke over these two months and significant increase of premature deaths were reported due to respiratory and cardiovascular diseases (Koplitz et al., 2016).

Despite the substantial environmental, socioeconomic and health impacts, the peatland fire emissions are still under-studied with respect to their chemical and physical properties. Thus, a mobile lab was deployed during the 2015 fire episode in Palangka Raya, Central Kalimantan, in order to obtain *in situ* ground based measurements of trace gases and aerosols directly from authentic peatland fire smoke. Samples discussed in this paper were collected from 18 peat plumes across six sites and were chemically speciated for ~90 gas phase species and ~70 particulate phase species. This paper focuses on the particulate phase chemistry, and a comprehensive description of gas phase emissions and optical properties is given in Stockwell et al. (2016).

Peatlands are globally distributed over ~400 Mha land area and hold ~550 MgC ha$^{-1}$ of carbon per 1 m depth, and they can be up to 20 m deep. It has been estimated that a total of ~5.4×10$^{14}$ kgC carbon is stored in peatlands, accounting for a significant fraction (44-71%) of the terrestrial carbon pool (Maltby and Immirzi, 1993; Yu et al., 2010). The majority of peatlands are in the cold boreal belt under ice, or maintained as wetlands or conserved areas and thus have escaped human interventions. However, tropical peatlands particularly in Malaysian and Indonesian lowlands are frequently converted to agricultural croplands, commercial forests, or pasture by draining the peatlands (Maltby and Immirzi, 1993). During 1996-1999 the Indonesian government excavated more than 4000 km of drainage channels throughout 1 Mha of peatland to cultivate rice under the former Mega Rice Project (MRP) (Page et al., 2009). After the project was abandoned in 1999, deforested and degraded peatlands were covered with secondary vegetation (Page et al., 2009). In recent decades, Indonesian peatland fires have occurred more frequently, intensively, and extensively. Degraded peatlands are at high risk of uncontrolled fire, because dry peat is highly combustible and secondary vegetation is more fire-prone

than the original forest (Langner and Siegert, 2009; Page et al., 2009; Page et al., 2002). Fires first occur in aboveground vegetation, then enter into the carbon-rich soils where they smolder and can spreadslowly beneath the surface until the peatland is flooded during the next monsoon (Page et al., 2009). The burned area does not easily regenerate to their primary landscape; instead, they are converted into ferns with patchy secondary vegetation that are prone to repeat fires (Chisholm et al., 2016).

Peat contains more than 85% organic matter by dry mass that is made of plant tissues at varying stages of decomposition, which major organic compound classes being cellulose, hemicellulose, lignin, cutine, humic acid and fulvic acid (Dehmer, 1995; Zulkifley et al., 2015; Dizman et al., 2015). Peat is categorized as fibric, hemic, or sapric based on the degree of decomposition. Fibric peat is the least degraded type with higher fiber content, while sapric peat is the most degraded peat type with an amorphous structure, and hemic peat has intermediate properties (Huat et al., 2011). Thus, peat soils carry biomarkers indicative of floral origin and these could be potentially used to identify peatland fire emissions. Levoglucosan, mannosan, syringaldehyde, vanillin, syringic acid, vanillic acid and n-alkanes are common biomass burning tracers and specific ratios of these compounds were suggested as indicators of peatland fire emissions in previous studies that analyzed the ambient air impacted by peat smoke (Fujii et al., 2014; Fujii et al., 2015a; Fujii et al., 2015b). Some organic compounds (e.g. PAHs) are highly enriched in peat smoke compared to raw peat biomass, showing over 100 times greater concentration in smoke than soil indicating their formation during combustion (Black et al., 2016).

Prior studies of peat burning emissions involved either laboratory experiments or collecting ambient aerosols at receptor sites impacted by peat smoke. Many of these studies primarily focused on

chemically characterizing gaseous emissions (Benner, 1977; Chen et al., 2007; Christian et al., 2003; Geron and Hays, 2013; May et al., 2014; McMahon et al., 1980; Ward, 1990; Hatch et al., 2015; Stockwell et al., 2015; Stockwell et al., 2014; George et al., 2016; Black et al., 2016; Iinuma et al., 2007; Yokelson et al., 1997) while fewer focused on the PM fraction (Black et al., 2016; Fujii et al., 2014; Fujii et al., 2015a; Iinuma et al., 2007). Peatland fire emissions were not considered in the biomass burning emission inventory published by Andreae and Merlet (2001). Akagi et al. (2011) updated this inventory to include peatland fires as a source of biomass burning emissions, but did not report an $EF_{PM2.5}$. Peat fire $PM_{2.5}$ emission factors reported in the literature have varied by a large scale, ranging from 5.9 g $kg^{-1}$ to 79 g $kg^{-1}$ with uncertainties associated with measurements of emissions of black carbon (BC) and organic carbon (OC) greater than 50% of the associated value (Black et al., 2016; Geron and Hays, 2013; Akagi et al., 2011). Thus, the global estimates of peat fire $PM_{2.5}$, OC, and BC emissions are associated with large uncertainties. The variation across lab-measured EF likely results from different burning conditions. In addition, the dissection of peat soil during sampling, handling, transport, and storage of peat can significantly alter its physical properties and subsequent combustion. Thus, *in situ* sampling of peat fire emissions under natural burning conditions is needed to accurately represent peat fire emissions in global peat fire emission estimates, parameterize human exposure studies, and climate and air quality models (Van der Werf et al., 2010; Page et al., 2002; Akagi et al., 2011).

The objectives of this paper are to characterize *in situ* peat PM emissions from different peat burning sites in Indonesia during the 2015 El Niño period, compute PM emission factors and develop source profiles for peat burning aerosols, and compare the peat PM emission factors from the literature

with our *in situ* measurements. This work is complementary to that of Stockwell et al. (2016) on the peat burning emissions of more than 90 gaseous species, brown carbon (BrC), black carbon (BC), and the mass absorption coefficients for the bulk OC due to BrC. Combined together, EFs for more than 150 gaseous and particulate species were determined, providing a wealth of chemical detail on these emissions and enabling the evaluation of the magnitude of $PM_{2.5}$ emissions and the ratio of particulate to gaseous carbon emitted from the 2015 El Niño peat fires.

## 2 Experimental details

### 2.1 Site description

A comprehensive description of sampling sites is given in Stockwell et al. (2016) and a brief overview is described here. $PM_{2.5}$ samples were collected from 18 separate plumes from 6 different peatland areas in Central Kalimantan, Indonesia from 1-7 November during the 2015 El Niño. The sites were carefully selected to represent different peat types (fibric, hemic, or sapric) and cover a range of burning depths ranging from 18 – 62 cm, averaging (±standard deviation) 34±12 cm. The sampled sites were located where the maximum fire activity is typically reported, in moderately to heavily disturbed areas by roads, canals and/or previous fires. The aboveground vegetation was nonexistent (due to previous fires) or limited to ferns or patchy secondary vegetation that was not burning in most cases. The samples were collected directly from visible plumes in smoldering peat. Sampling was immediately stopped during any occasional flaming combustion events within aboveground vegetation in the vicinity to ensure sampling of pure smoldering peat emissions.

Each plume was identified by an English letter (E-Z to AA) and the complete description of the plumes including peat type, burning depth, and surface fuel is given in Table S1 Stockwell et al. (2016). Duplicate samples were collected from plumes E, F and W, bringing the total number of PM samples to 21. Because of the variability across duplicate sample, both values were used in calculating study-averages. Plume Y showed a different emission profile from the others likely due to co-burning of leaf litter at this shallow peat burning site. Thus, plume Y was excluded from average calculations but individual values are reported in Table S1 and corresponding figures.

### 2.2 Sample collection

A comprehensive description of sample collection is given in Stockwell et al. (2016). In brief, $PM_{2.5}$ was collected using a custom-built, two-channel PM sampler. The inlet was positioned at a point where the plume of smoke cooled to near-ambient temperature, to allow for gas-particle partitioning to equilibrate prior to sample collection. The sample inlet was not fixed to a point and always followed the plume path when the plume direction changed due to wind. The PM was collected on pre-cleaned 47 mm quartz fiber filters (QFF) and pre-weighed Teflon filters (PALL, Life Sciences, Port Washington, NY) preceded by two 2.5 µm sharp-cut cyclones (URG). The filtered air was then passed to the land-based Fourier transform infrared (LA-FTIR) spectrometer multipass cell for the measurement of gas phase species as described by Stockwell et al. (2016). Sampled filters were stored in the dark and frozen (-20 °C) and were shipped frozen to the University of Iowa for chemical analysis.

Field blanks were collected for every fifth sample. For some samples a second (backup) QFF filter was placed in series behind the first (front) QFF filter in order to assess the positive sampling artifacts from carbonaceous gas adsorption. Filter samples were collected upwind of the plumes for ~20 minutes (similar to smoke sampling duration) in order to account for background $PM_{2.5}$.

### 2.3 $PM_{2.5}$ mass, elemental carbon and organic carbon measurement

A complete description of PM mass, elemental carbon (EC) and organic carbon (OC) measurements are given in Stockwell et al. (2016). In brief, PM mass was calculated as the difference of pre-and post-sampling filter weights of Teflon filters after conditioning for 48 hours in a desiccator. The relative error in the PM mass measurements was propagated from the standard deviation of the triplicate

measurements of pre-and post-sampling filter weights, the standard deviation of background PM masses, and 10% of the PM mass concentration, which is a conservative estimate of the analytical uncertainty associated with the mass measurement. Ambient background $PM_{2.5}$ concentrations were very similar across all the sites and on average the ambient $PM_{2.5}$ contributed only 0.60% of the sampled $PM_{2.5}$ mass, indicating that the ambient PM contribution was very small compared to PM concentration in the peat smoke. Nevertheless, the average background concentration was subtracted from the sample concentrations in order to calculate pure peat fire emissions.

EC and OC were measured by thermal optical analysis following the NIOSH 5040 method using 1.00 $cm^2$ punches of quartz fiber filters (Sunset Laboratories, Forest Grove, OR) (NIOSH, 2003). The uncertainty in OC measurements was propagated from the standard deviation of the background filters, the standard deviation of the back-up filters, and 10% of the OC concentration, a conservative estimate of the method precision in replicate measurements (NIOSH, 2003). The uncertainty of EC measurements was propagated from the instrumental uncertainty (0.05 µg $cm^{-2}$), 5% of the measured EC, and 5% of pyrolyzed carbon, which refers to organic carbon that charred during analysis.

## 2.4 Water-soluble organic carbon

A 1.053 $cm^2$ sub-sample of QFF filter was analyzed for water soluble organic carbon (WSOC) using a total organic carbon analyzer (GE, Sievers 5310 C). WSOC was extracted into 15.0 mL of >18.2 MΩ resistivity ultra-pure water (Thermo, Barnstead Easypure II) using acid washed (10% nitric acid) and pre-baked (550 °C for 5.5 hours) glassware. Inorganic carbon was removed with an inorganic carbon remover (GE, Sievers ICR). WSOC was measured in triplicate and quantified using standard

calibration curves prepared from potassium hydrogen phthalate (Ultra Scientific). The WSOC concentration in the sampled plumes was calculated using the extraction volume, total filter area, and sampled air volume. The uncertainty of the WSOC measurement was propagated using the standard deviation of the triplicate measurements, standard deviation of the background filters and 10% of the WSOC concentration. The fraction of water-insoluble organic carbon (WIOC) was calculated by subtracting the WSOC concentration from total OC concentration. The error of WIOC concentration was propagated from individual uncertainties of OC and WSOC.

## 2.5 Water-soluble inorganic ions

Water-soluble inorganic ions were quantified in aqueous extracts of Teflon filters by ion exchange chromatography coupled with conductivity detection as described in detail elsewhere (Jayarathne et al., 2014). In brief, half of the Teflon filter was uniformly wet with 50 µL of isopropyl alcohol and subsequently extracted into 15.0 mL ultra-pure water (>18.2 MΩ resistivity) by shaking 12 hours at 125 rpm. For cation analysis, a Dionex IonPac CS12A column was used with the mobile phase of 20 mM methane sulfonic acid at 0.5 mL $min^{-1}$ flow rate. A Dionex IonPac AS22 anion column with the mobile phase of 4.5 mM sodium carbonate ($Na_2CO_3$) and 1.4 mM sodium bicarbonate ($NaHCO_3$) at a flow rate of 1.2 mL $min^{-1}$ was used for anion separation. A conductivity detector (Thermo) was used for detection and was preceded by a self-regenerating suppressor, CERS-500 and AERS-500 for cations and anions, respectively.

## 2.6 Total metals

Teflon filters were cut in half using ceramic blades and then digested in mixture of 2:1 concentrated nitric and hydrochloric acid (TraceMetal Grade, Fisher Chemical) using a MARS 6 microwave assisted digestion system (CEM Corporation, Matthews, NC) at 200 °C for 13 minutes following US EPA Method 3052 (USEPA, 1995). Extracts were filtered (0.45 µm PTFE) and analyzed for metals using a Thermo X-Series II quadrupole ICP-MS instrument (Thermo Fisher Scientific Inc., Waltham, MA, USA) (Peate et al., 2010). The instrument was calibrated against IV-ICPMS-71A ICP-MS standard (Inorganic Ventures) at concentrations ranging from 0.1 - 50 ppb. The metal concentration in the extract was converted to metal oxide concentration in the sampled plumes (µg $m^{-3}$) using extraction volume, total filter area, sampled air volume, metal to metal oxide mass ratio, and the natural metal isotope abundance (Rosman and Taylor, 1999). The uncertainty of the measurement was propagated using the method detection limits, standard deviation of the field blank filters and 10% of the metal concentration.

## 2.7 Organic species

Organic species were quantified in organic extracts of QFF by gas chromatography mass spectrometry (GC-MS) as described in detail elsewhere (Al-Naiema et al., 2015). In brief, quartz fiber filters were sub-sampled to obtain ~200 µg C prior to organic species characterization. These sub-samples were spiked with deuterated internal standards which were used in quantification: pyrene-$D_{10}$, benz(a)anthracene-$D_{12}$, cholestane-$D_4$, pentadecane-$D_{32}$, eicosane-$D_{42}$, tetracosane-$D_{50}$, triacontane-$D_{62}$, dotriacontane-$D_{66}$, hexatriacontane-$D_{74}$ , levoglucosan-$^{13}C_6$, and cholesterol-$D_6$. Each sub-sample was then stepwise extracted in 2×20 mL aliquots of hexane followed by 2×20 mL aliquots of acetone by

ultra-sonication (60 sonics $min^{-1}$, 5510-Branson) for 15 minutes. The solvent extracts were subsequently concentrated to a final volume of ~100 µL using Turbovap (Caliper Life Sciences, Turbo Vap LV Evaporator) and micro-scale nitrogen evaporation system (Thermo Scientific, Reacti-Vap™ Evaporator) upon high-purity nitrogen (PRAXAIR Inc.). These extracted samples were stored at -20 °C until the chemical analysis.

Organic species in filter extracts were quantified using gas chromatography coupled to mass spectrometry (Agilent Technologies GC-MS 7890A) equipped with an Agilent DB-5 column (30 m × 0.25 mm × 0.25 µm) with electron ionization (EI) source using a temperature range from 60 to 300 °C. Helium was utilized as the carrier gas, and the 3 µL aliquots of the extracts were injected in splitless mode. More oxygenated polar compounds were analyzed following trimethylsilyl (TMS) derivatization (Stone et al., 2012). Briefly, 10 µL of the extract was blown down to complete dryness and reconstituted in 10 µL of pyridine (Burdick & Jackson, Anhydrous). A 20 µL of the silylation agent N,O-bis-(trimethylsilyl)trifluoroacetamide (Fluka Analytical, 99%) was added to the mixture, and was heated for 3 hours at 70 °C to complete the silylation reaction. The silylated samples were immediately analyzed for polar compounds.

Responses of analytes were normalized to the corresponding isotopically-labeled internal standard and five-point linear calibration curves (with correlation coefficients, $R^2 \geq 0.995$) were utilized for the quantification of organic species. Compounds that were not in the standards were measured by assessing the response curve from the compound that was most analogous in structure and retention time. The analyte concentration in the extract was converted to ambient concentrations (µg $m^{-3}$) using extraction volume, the total filter area, and sampled air volume. The analytical uncertainties for the

measured species were propagated from the method detection limits, standard deviation of the field blank filters and 20% of the measured concentration, which is based upon the spike recoveries of individual species being allowed to vary within 100±20%.

## 2.8 Emission factor calculation

The mixing ratios of $CO_2$, CO, $CH_4$ and ~90 other gases were quantified by a field-deployed Fourier transform infrared (FTIR) spectrometer combined with whole air sampling (WAS) (Stockwell et al., 2016). The carbon mass balance approach was used to determine fuel-based emission factors (EF) for gases, in units of mass of analyte per kilogram of fuel burned (g $kg^{-1}$) (Stockwell et al., 2016). Carbon monoxide was used as the reference species to calculate the EF of particulate species. For this purpose, carbon monoxide mass drawn through the filter ($M_{CO}$) that was measured in series by FTIR, the mass of the analyte ($M_X$; i.e., PM mass, EC, OC), and emission factor of carbon monoxide ($EF_{CO}$) were used to calculate the emission factors of the desired analyte ($EF_X$) using equation 1.

$$EF_X = \frac{M_X}{M_{CO}} \times EF_{CO} \quad (1)$$

Uncertainty in $EF_X$ was propagated from the relative uncertainty of $EF_{CO}$, conservatively estimated as 5% of the value and the analytical uncertainty of the considered analyte.

## 2.9 Modified combustion efficiency

The modified combustion efficiency (MCE) was calculated as $MCE = \Delta CO_2/(\Delta CO + \Delta CO_2)$ and was used as an indicator of flaming combustion (MCE > 0.9) and smoldering combustion (~0.72-

0.84) (Yokelson et al., 1996). Notably, the filter-integrated MCE values reported herein correspond to the duration of filter sample collection and could differ slightly from those reported by Stockwell et al. (2016) that included additional measurements.

## 3 Results and discussion

### 3.1 Emission of $PM_{2.5}$

$EF_{PM2.5}$ for *in situ* Indonesian peat burning ranged from 6.04– 29.6 g $kg^{-1}$ for 18 plumes, averaging (± standard deviation) 17.3±6.0 g $kg^{-1}$ (Figure 1). The percent difference across duplicate samples was 57%, 37%, and 8% for plumes E, F, and W, respectively, indicating temporal variability in emissions from a single plume as the peat fire progresses. This in-plume variability in the field is likely to result from the spread and progression of the fire, consistent with peat samples burned batch-wise in laboratory settings that show $EF_{PM2.5}$ decreases on the time scale of hours during combustion (Black et al., 2016). The overall relative standard deviation (RSD) of $EF_{PM2.5}$ in this study was 35%, indicating that variability across plumes is on the same order as in-plume variability.

The average $EF_{PM2.5}$ for Indonesian peat burning is within the range of values reported in prior laboratory studies (6 – 66 g $kg^{-1}$; Table 1). Generally, the variability attributed to natural variation within the fuel, including its chemical composition (e.g., C-content), moisture content, and fuel density; and burn conditions (e.g., extent of flaming versus smoldering combustion) (Iinuma et al., 2007).The average $EF_{PM2.5}$ reported by Black et al. (2016) for two sources of North Carolina peat (7.1±5.6 g $kg^{-1}$ and 5.9±6.7 g $kg^{-1}$) are in the lower range of $EF_{PM2.5}$ observed in this study. The peat fires studied by Black et al. (2016) exhibited higher MCE values (0.80 – 0.88) compared to this study (0.73 – 0.83), in

which the former corresponds to lower PM emissions (McMeeking et al., 2009) and may have resulted from oven-drying their peat samples prior to combustion. Meanwhile, the observed $EF_{PM2.5}$ value for *in situ* Indonesian peat burning is lower than the $EF_{PM2.5}$ values reported by other laboratory studies: 46±21 g kg$^{-1}$ by Geron and Hays (2013); 33-44 g kg$^{-1}$ (for $PM_{10}$) by Iinuma et al. (2007); 42 g kg$^{-1}$ by Chen et al. (2007); 35 g kg$^{-1}$ by May et al. (2014) and 30±20 g kg$^{-1}$ by McMahon et al. (1980). These higher $EF_{PM2.5}$ could be due to natural variability in the peat composition, and/or experimental variables such as sampling from early stage of fires or extent of dilution. We also cannot rule out that the smoke in some previous laboratory studies was concentrated enough to increase gas-particle partitioning beyond the level in our samples. Further, alterations to peat between the field and laboratory may have an effect through the transporting and handling of peat soils; differences associated with igniting the peat sample (e.g. heated coil vs propane torch); the edge effect due to igniting small chunk of peat; and sustainability of the fire during the time of sample collection could also affect the $EF_{PM2.5}$. Because the $EF_{PM2.5}$ computed during this study correspond to natural conditions of peat burning that was not handled, transported or processed disturbing the peat soil micro-properties, the reported measurements are not subject to potential fuel alteration.

### 3.2 Emission of OC, EC, and WSOC

Across the studied plumes, $EF_{OC}$ ranged from 1.76 – 26.9 g kg$^{-1}$, averaging 12.4±5.4 g kg$^{-1}$ (Figure 2). The high OC mass fraction of PM (72±11 %) is in a good agreement with literature reported values 73-89 % by Black et al. (2016) and 94% by Chen et al. (2007) for $PM_{2.5}$ from peat combustion in laboratory studies. The $EF_{EC}$ ranged from 0.09 – 0.44 g kg$^{-1}$, averaging 0.24±0.10 g kg$^{-1}$ (Table 2). The high $EF_{OC}$ and low $EF_{EC}$ values are consistent with purely smoldering combustion with MCE values of

0.725 - 0.833 as discussed by Stockwell et al. (2016). The optically measured $EF_{BC}$ in $PM_1$ by photoacoustic extinctiometry (PAX) (0.006±0.002 g $kg^{-1}$) was noticeably lower than that of filter based $EF_{EC}$ likely due to sampling of char particles by filters, different PM size cuts, and differences in measurement methods (Stockwell et al., 2016). Overall, both optical and chemical measurement methods employed in Indonesia and prior studies of EC in peat burning emissions (Table 1) agree that $EF_{EC}$ and $EF_{BC}$ are very small compared to $EF_{OC}$.

The OC:EC ratio for *in situ* Indonesian peat burning ranged from 27-129, averaging 67±26. This is in the middle of the range of OC:EC values reported previously for peat combustion (Table 1). The PAX results showed that the ratio of light absorption at 405 nm relative to 870 nm wavelength was approximately 50 (Stockwell et al., 2016), whereas a ratio close to 2.2 is indicative of absorption by pure BC (Bond and Bergstrom, 2006). Thus, the light absorption by peat smoke is largely due to BrC and the measured high BrC:BC absorption ratio (52) is similar to the measured OC:EC ratio (Stockwell et al., 2016). The bright yellow color of the PM collected filters (Figure S1) is also an indication of the light-absorbing nature of the OC and a very small relative emission of EC.

On average, only a minor fraction of OC was water soluble (16±11 %) and the majority (84±11 %) was water insoluble (Table 2). Hence, the majority of OC is composed of hydrophobic organic compounds. These results are consistent with prior observations of high relative concentrations of aliphatic organic species in peat and peat-burning aerosol reported previously (Iinuma et al., 2007; McMahon et al., 1980). The low water-solubility and presence of hydrophobic organic species likely contribute to the hydrophobicity and low CCN activity of fresh peat burning emissions (Dusek et al., 2005).

### 3.3 Chemical composition of $PM_{2.5}$

OC accounted for the major fraction of $PM_{2.5}$ (72±11 %) while EC was detected only in 15 plumes and on average comprised 1.2 % of $PM_{2.5}$ (Table 1). Minor contributions to $PM_{2.5}$ were observed for water-soluble ions (1.2%) and metal oxides (less than 0.1 %) (Table 2). The sum of OC, EC, water-soluble ion and metal oxide masses comprised, on average, 74±11 % of gravimetrically measured $PM_{2.5}$ mass.

The remaining $PM_{2.5}$ mass is expected to be primarily from elements associated with carbon in forming organic matter (e.g., O, H, N). Assuming that no major chemical species were unmeasured, we estimate organic matter (OM) as the difference between $PM_{2.5}$ mass and the sum of EC, water-soluble ions, and metal oxides (OM = $PM_{2.5}$ – [EC+ions+metals oxides]). The linear regression analysis of this estimate of OM and measured OC correlated strongly ($R^2$ = 0.93) indicating their dependent co-variance (Figure 2). The slope of the regression line is 1.26±0.04 OM $OC^{-1}$ and provides the conversion factor of OC to OM for fresh peat burning aerosols. This OC to OM factor is in the range of values typically observed for gasoline combustion (1.1-1.3) (Schauer et al., 2002, 1999) and below those used for biomass burning (1.4-1.8) (Reid et al., 2005), which is expected to result from the semi-fossilized nature of the peat fuel and the water-insoluble (section 3.2) and aliphatic-rich (section 3.5) nature of OC.

### 3.4 MCE

The calculated MCEs were indicative of smoldering combustions with values ranging 0.725-0.833 (average = 0.78±0.04) (Yokelson et al., 1996). Burn depth and MCE were negatively correlated (r

= -0.738; p = 0.001; Figure S2) consistent with higher emission of $CO_{(g)}$ relative to $CO_{2(g)}$ for deep peat combustion, potentially due to less oxygen supply. Over the small range of observed MCEs and purely smoldering combustion, neither MCE nor burn depth were correlated with PM mass, EC, or OC emission factors (p > 0.23) and thus, did not noticeably affect PM emissions.

### 3.5 Organic species

A subset of samples (n = 10), representing at least 1 sample per sample collection site was analyzed for anhydrosugars, lignin decomposition compounds, alkanes, hopanes, PAHs and sterols. On average, the quantified organic compounds accounted for ~9 % of the total OC mass on carbon basis with major contribution from alkanes (6.2 %), followed by anhydrosugars (2.1 %), lignin decomposition products (0.36 %), hopanes (0.12 %), sterols (0.06 %) and PAHs (0.03%) (Figure 3). Up to approximately 5% more of the OC is expected to come from n-alkenes, some oxy-PAH, additional lignin decomposition products, nitrophenols that were measured in peat emissions by Iinuma et al. (2007). The remaining OC remains unresolved and is likely to include isomers of the abovementioned compounds (e.g., branched alkanes) and high-molecular weight organic compounds. Plume Y that was obtained from shallow peat burning sites with plant roots observed in the burn pit had a different emission profile with a larger contribution from anhydrosugars (16 %) compared to lignin decomposition products (2.8 %) and alkanes (1.6 %). Plume Y thus represents the co-burning of peat with surface vegetation and was excluded from average calculations that represent sub-surface burning of peat. The full emission profiles for each individual plume is reported in Table S1.

### 3.5.1 Alkanes

The homologous series of n-alkanes and select branched alkanes were quantified in emissions from Indonesian peat burning. The *n*-alkanes with carbon numbers ranging $C_{18}$-$C_{34}$ were detected in all samples analyzed by GCMS, with higher-carbon number homologs observed in many samples (Table S1). The *n*-alkane emission factor ($EF_{alk}$) for the quantified species ranged 456-3834 mg $kg^{-1}$ (Table S1).

On average, *n*-alkanes contributed 6.2% of OC mass. This OC mass fraction is consistent with results from Iinuma et al. (2007) for Indonesian and German peat burning and is remarkably higher than other types of biomass burning OC for which this OC fraction is typically less than 1% (Hays et al., 2002; Iinuma et al., 2007). The high *n*-alkane contribution to OC results from the high lipid content of peat that accumulates from plant waxes (e.g. cutin, suberin) during decomposition (Ficken et al., 1998). The *in situ* source emissions and prior measurements of peat combustion in the field (Fujii et al., 2015a) and in the laboratory (Iinuma et al., 2007)agree that n-alkanes can be used to distinguish peat emissions from other types of biomass burning and other combustion sources by their high contribution to particle-phase OC.

The most abundant *n*-alkane ($C_{max}$) was consistently observed for the $C_{31}$ carbon homolog (Table S1). This is the same $C_{max}$ value observed by Iinuma et al. (2007) for Indonesian peat, while in ambient air impacted by Indonesian peat burning, Fujii et al. (2015a) and Abas et al. (2004) reported $C_{max}$ at $C_{27}$. This variability in $C_{max}$ likely derives from in the peat material, but may be influenced by atmospheric aging as the differences in $C_{max}$ are aligned with fresh and aged peat burning aerosol.

As shown in Figure 4, *n*-alkanes demonstrated a slight odd carbon preference (Figure 4) that is indicative of biogenic material, particularly plant waxes (Fine et al., 2002; Oros and Simoneit, 2001a, b;

Baker, 1982). The carbon preference index (CPI) was calculated using concentrations of $C_{24-32}$ n-alkanes following Fujii et al. (2015a) and ranged 1.22-1.60, averaging 1.42±0.10. Comparable CPI values have been reported previously for laboratory emissions from peat collected in Indonesia (1.5), Germany (1.8) (Iinuma et al., 2007), and North Carolina (1.4-1.5) (George et al., 2016). These CPI values are low in comparison to emissions from foliage, softwood, and hardwood combustion emissions that range 1.6-6.2 (Hays et al., 2002; Yamamoto et al., 2013). Together, the high *n*-alkane mass fraction and CPI values of 1.4± 0.2 and are characteristic features Indonesian peat fire emissions.

### 3.5.2 Anhydrosugars

Pyrolysis of cellulose and hemicellulose generates anhydrosugars, of which levoglucosan, mannosan and galactosan were quantified. Anhydrosugar EF ($EF_{anh}$) ranged 157-2041 mg $kg^{-1}$ and averaged 543±598 mg $kg^{-1}$. The dominant anhydrosugar was levoglucosan (averaging 46±40 mg $gOC^{-1}$), followed by mannosan (0.93±0.76 mg $gOC^{-1}$) and galactosan (0.14±1.13 mg $gOC^{-1}$) (Figure 5, Table 2). Levoglucosan was the most abundant individual species quantified and contributed 0.3-6.0% of OC mass (Table S1). A significant correlation was not observed between $EF_{OC}$ and $EF_{levoglucosan}$ ($p = 0.4$) in contrast to Sullivan et al. (2008) who observed the correlation of these values for biomass burning emissionsfrom grass, duff, chaparral, softwood and hardwood fuels ($R^2$=0.68) . The variable cellulose content across peat soils likely contributes to this lack of correlation.

While relative ratios of levoglucosan, mannosan, and galactosan have been used to distinguish between various types of biomass combustion emissions (Engling et al., 2014), peat burning emissions did not exhibit consistent ratios of these species. The levoglucosan to mannosan ratio ranged widely 27-

160 with an average (± standard deviation) of 55±41. Meanwhile, Iinuma et al. (2007) reported this ratio to be 11 and Fujii et al. (2015a) reported it to average 15. Because of the variability across studies and the expected dependence of this ratio on biomass cellulose content and composition (Sullivan et al., 2008), this ratio is insufficient to distinguish peat combustion from other biomass types.

### 3.5.3 Lignin decomposition compounds

Syringaldehyde (S), vanillin (V), syringic acid (SA) and vanillic acid (VA) derived from lignin pyrolysis were quantified as lignin decomposition products, with a combined EF ranging 15-154 mg $kg^{-1}$ and averaging 80±50 mg $kg^{-1}$ (Table S1). Correlations among aldehydes (V and S) were not significant, possibly due to V partitioning to the gas phase, as indicated by its detection on backup filters while others species (S, VA and SA) were detected only on front filters. We observed a moderate significant correlation ($R^2$=0.65; p=0.004) between $EF_{VA}$ and $EF_{SA}$. Based on linear regression analysis, 1.9±0.2 was determined as the ratio of VA:SA for freshly emitted peat smoke (Figure 7). Because of its consistency, the VA:SA ratio is recommended as an indicator of peat smoke. Previously, VA to SA ratio has been suggested as an indicator for peat fire emissions because ambient aerosols affected by Indonesian peat fires showed a VA:SA ratio of 1.7±0.36, while the unaffected aerosols had a ratio of 0.59±0.27 (Fujii et al., 2015a).

### 3.5.4 PAHs, hopanes and sterols

Polycyclic aromatic hydrocarbons (PAHs) were observed in emissions from Indonesian peat burning and the 18 PAHs that were quantified are listed in Table 2. For the measured species, $EF_{PAH}$

ranged 1.7-17 mg kg$^{-1}$ and were consistent with previously reported $EF_{PAH}$ values, 6-25 mg kg$^{-1}$ for laboratory peat burning studies (Black et al., 2016; Iinuma et al., 2007). PAH composition was dominated by pyrene, chrysene, methylfluoranthene, fluoranthene, and retene, which accounted for ~56% of the measured PAH emissions (Table 2). Several biomass burning studies have reported retene, a biomarker of softwood combustion, as the most abundant PAH in wood smoke (Fine et al., 2002; Hays et al., 2002; Schauer and Cass, 2000), whereas it contributed only 8% of the measured PAH in this study.

Benz(a)anthracene, benzo(a)pyrene, benzo(b)fluoranthene, benzo(k)fluoranthene, chrysene, and dibenz(a,h)anthracene, which are categorized as probable human carcinogens by the US Environmental Protection Agency (USEPA, 2008), were detected in peat burning aerosols and together these PAHs accounted for 39% of total quantified PAH species. The toxic equivalency factor was estimated for quantified PAHs to estimate the overall human health hazard level (Nisbet and LaGoy, 1992). The estimated B[a]P equivalent toxicity value ranged 0.05-0.39 B[a]P eqs, mg kg$^{-1}$, averaging 0.13±0.10 B[a]P eqs, mg kg$^{-1}$ and comparable to previously reported toxicity values for peat smoke, 0.12-0.16 by Black et al, (2016). The total PAH concentration in undiluted peat smoke ranged 0.3-18 μg m$^{-3}$ and was similar to PAH concentrations reported for exhaust smoke of a coke-oven (25 μg m$^{-3}$), aluminum smelting (15 μg m$^{-3}$), diesel engines (5 μg m$^{-3}$), and gasoline engines (3 μg m$^{-3}$) (Khalili et al., 1995; Armstrong et al., 2004).

To the best of our knowledge, hopanes have not been previously quantified in peat fire emissions. 17α(H)-22,29,30-Trisnorhopane, 17β(H)-21α (H)-30-norhopane, and 17α(H)-21β(H)-hopane were identified using authentic standards and quantified in pure peat smoke for the first time. $EF_{hopanes}$

ranged 11-37 mg $kg^{-1}$, averaging 17±8 mg $kg^{-1}$ (Table S1). Terpenoid and hopanoid hydrocarbon compounds that have the hopane-skeleton are ubiquitous in peat soils (Ries-Kautt and Albrecht, 1989; Venkatesan et al., 1986; Quirk et al., 1984; López-Días et al., 2010; Del Rio et al., 1992; Dehmer, 1995). Thus, presence of hopanes in peat smoke is not unexpected. Norhopane had the highest OC mass fraction followed by trisnorhopane and hopane (Table 2). A fairly consistent ratio of 0.25:0.60:0.15 was observed among trisnorhopane, norhopane, and hopane irrespective of the sampling site and burning depth, indicating the formation of hopanes are independent of burning conditions (Figure S3). The observed hopanes ratio is clearly distinct from that of diesel (0.04:48:48) (Schauer et al., 1999) and noncatalyst gasoline (0.10:0.42:0.48) (Schauer et al., 2002) engine emissions. However, it is comparable to the hopane ratio of lignite (0.23:0.66:0.11) and sub-bituminous (0.29:0.49:0.22) coal smoke (Oros and Simoneit, 2000). This indicates similarities of terpenoid and hopanoid hydrocarbons in peat soils and coal deposits and these are younger on the geological timescale than crude oil.

Stigmasterol, β-sitosterol, and campesterol were detected in peat smoke and accounted for 0.14-1.7 mg $gOC^{-1}$ of OC mass fraction (Table S1). Sterols have been identified in peat soils with a major contribution from β-sitosterol (Del Rio et al., 1992; López-Días et al., 2010). Similarly, β-sitosterol is the predominant sterol in PM (Table 2), indicating the emission of peat constituents to the atmosphere as PM during smoldering.

### 3.6 Water-soluble inorganic ions

Water-soluble ions accounted for only 1.1% of the PM mass and total quantified $EF_{ions}$ ranged 45 – 490 mg $kg^{-1}$, averaging 201±144 mg $kg^{-1}$. Ammonium and chloride were detected in all the

samples with average EFs of 92±61 mg $kg^{-1}$ and 75±52 mg $kg^{-1}$, respectively. Frequency of detection (FOD) for sulfate, nitrate, and fluoride was 83%, 61%, and 56% and EFs ranged 2-133 mg $kg^{-1}$, 0.2-6.8 mg $kg^{-1}$, and 0.4-45.9 mg $kg^{-1}$, respectively. PM mass fractions of ammonium vs sulfate (r = 0.95, p<0.001) and ammonium vs chloride (r = 0.89, p<0.001) were strongly correlated indicating that a major fraction of inorganics in PM is in the form of $(NH_4)_2SO_4$ and $NH_4Cl$. The molar concentration of gaseous $NH_3$ and NO+HONO were 33 times and 312 times higher than that of $NH_4^+$ and $NO_3^-$, respectively consistent with a dominance of gas phase precursors in fresh peat burning emissions (Stockwell et al., 2016). The atmospheric oxidation of NO and HONO could increase the concentration of $NO_3^-$ (Gankanda and Grassian, 2013; Gankanda et al., 2016), while acid base reactions convert $NH_3$ to $NH_4^+$, thus leading to increased concentrations of these secondary inorganic products in aged peat smoke.

Potassium has been used as an indicator of biomass burning, both on its own and in concert with levoglucosan (Simoneit et al., 1999; Sullivan et al., 2008; Chuang et al., 2013; Gao et al., 2003). From peat smoldering fires, extremely low potassium emissions (0.03% of PM mass) were observed, at concentrations too low to be a useful indicator species as described by Sullivan et al. (2014) and Fujii et al. (Fujii et al., 2015a).

### 3.7 Metals

Metal oxides accounted for only 0.1% of the PM mass and their EF ranged 7 – 24 µg $kg^{-1}$, averaging 13±5 µg $kg^{-1}$ (Table 2, Table S1). The metal fraction was dominated by Al, Ti, V, Mn, Ni, Sr, and Ba which are commonly found in peat soil (Dizman et al., 2015). The lower $EF_{metal}$ values relative

to other quantified species (i.e. OC) indicate the minimum influence of re-suspended soil dust to $PM_{2.5}$. Further, combustion at temperatures lower than 400 $^oC$, indicative of smoldering conditions, precludes metal transfer to the aerosol phase (Raison et al., 1985; Aswin et al., 2004).

### 3.8 Emission estimates from 2015 Indonesian peat fires

The emissions from Indonesian peat fires during the 2015 El Niño were estimated using mean EFs calculated in this study for an estimated burned area of $8.5 \times 10^5$ ha (Whitburn et al., 2016), an average burning depth of 34±12 cm (calculated during this study (Stockwell et al., 2016)), and a peat bulk density of 0.120±0.005 g $cm^{-3}$ (Konecny et al., 2016). The uncertainty of the estimated value is propagated using standard deviation of the mean EFs, burn depth, and peat bulk density. However, the uncertainty of burned area is not defined.

In this way, the total $PM_{2.5}$ released to the atmosphere from this fire event was estimated to be 3.2 - 11 Tg, averaging 6.0±5.5 Tg with major contribution from OC (4.3 Tg) followed by EC (0.08 Tg) and water-soluble ions (0.07 Tg) (Table 3). Combining our OC and EC emission factors with gas-phase EFs of $CO_2$, CO, $CH_4$, and other carbon containing gases from Stockwell et al. (2016), we estimate a total carbon emission of 205±77 TgC to the atmosphere, of which 73% was as $CO_2$ (149±71 TgC), 21% as CO (44±30 TgC), 1.2% as $CH_4$ (2.5±2.6 TgC), 2.7% as other carbon containing gases (5.5±1.3 TgC), 2.1% as OC (4.3±4.3 TgC) and 0.04% as EC (0.083±0.081 TgC). Our carbon emission estimates are in good agreement with Huijnen et al. (2016) who estimated total C emissions of 227±67 TgC for this fire event. However, this is ~8 times lower than the carbon emissions estimated for the 1997-98 Indonesian peat fires (810-2570 TgC) (Page et al., 2002).

### 3.9 Conclusions

$PM_{2.5}$ was collected from authentic *in situ* peat smoke during the 2015 El Niño peat fire episode in Central Kalimantan, Indonesia and was chemically characterized for PM mass, EC, OC, water-soluble ions, metals, and organic species. Fuel based $EF_{PM2.5}$ ranged from 6.0 - 29.6 g $kg^{-1}$ averaging 17.3±6.0 g $kg^{-1}$ and we estimate 3.2 - 11 Tg of $PM_{2.5}$ were released to the atmosphere during the 2015 El Niño peat fire episode. OC accounted for the major fraction of PM mass while EC, water-soluble ions, and metal oxides comprised only a minor fraction of PM mass. Combining our $EF_{OC}$ and $EF_{EC}$ with gas-phase EFs of $CO_2$, CO, $CH_4$, and other carbon containing gases from Stockwell et al. (2016), we estimate a total carbon emission of 205±77 TgC to the atmosphere. OC and EC comprised 2.1% and 0.04% of total carbon emissions, respectively.

Overall, chemical speciation of OC revealed the following characteristics of peat burning emissions: high OC mass fractions (72%), primarily water-insoluble OC (84±11% C), low EC mass fractions (1%), vanillic to syringic acid ratios of 1.9, and relatively high n-alkane contributions to OC (6.2% C) with odd carbon preference CPI (1.2-1.6). This chemical profile is in good agreement with prior studies of Indonesian peat burning using laboratory measurements (Christian et al., 2003; Iinuma et al., 2007) and ambient aerosol studies in Indonesia (Fujii et al., 2015a; Fujii et al., 2015b) as well as laboratory studies of peat emissions from other locations (Black et al., 2016; Geron and Hays, 2013; Chen et al., 2007). The similarities of the peat burning chemical profiles determined in this *in situ* emissions characterization and prior and laboratory studies reveal that laboratory studies can accurately capture the fractional composition of PM and OC. However, greater discrepancies arise in the absolute

$EF_{PM2.5}$ emissions (Table 1) across field and laboratory studies, with the former typically yielding lower $EF_{PM2.5}$ values. The absolute differences in $EF_{PM}$ are expected to result from several factors, such as fuel composition and moisture content, combustion conditions, and timing of PM sampling.

Knowledge of chemical characteristics of peat emissions can be used in source identification and apportionment modeling at a receptor site is impacted by peatland fire emissions. Further, they can allow for assessment of acute and chronic hazards associated with exposures to high concentrations of PM and PAH from peat smoke during the fire season (Armstrong et al., 2004; Kim et al., 2013).

The quantitative emission factors developed in this study for Indonesian peat burning are the most representative of natural peat burning conditions and may be used to update regional/global emission inventories which are currently based on EFs computed from laboratory studies. The most recent emission inventory compiled by Akagi et al., (2011) does not include an EF value for $PM_{2.5}$ for peatl fire emissions, and the reported $EF_{OC}$ and $EF_{EC}$ correspond to peatland burning that include an estimate of emissions of above-ground tropical forest with peat. Further, the $EF_{OC}$ reported in Akagi et al. (2011) is 50% lower than the average $EF_{OC}$ observed in this study, which would underestimate the $PM_{2.5}$ OC emissions observed in the field. Thus, the use of these *in situ* EFs in updates to emission inventories can provide more accurate emission estimates. Further, more studies should be carried out downwind to evaluate the effects of atmospheric dilution and atmospheric photochemical reactions on the chemical composition of peat fire PM.

**Acknowledgements.** This research was primarily supported by NASA Grant NNX13AP46G to SDSU and UM. The research was also supported by NASA grant NNX14AP45G to UM. We also

acknowledge the T. Anne Cleary International Dissertation Research Fellowship awarded by the Graduate College, University of Iowa and Center for Global and Reginal Environmental Research (CGRER) graduate student travel award for field research. We also thank Dr. David Peate, Iowa Trace Element Analysis Laboratory for the assistance given during metal analysis. We are also grateful to Laura Graham, Grahame Applegate and the BOS field team for their excellent support during the sample collection.

**Figure 1:** Emission factors of $PM_{2.5}$, EC, OC, water-soluble ions, and metal oxides for the average and individual peat smoke plumes. Error bars represent one standard deviation of the average or the propagated analytical uncertainty. $EF_{PM2.5}$ was dominated by OC (72%) with minor contributions from EC (<1%), ions (<1%) and metal oxides (<0.1%).

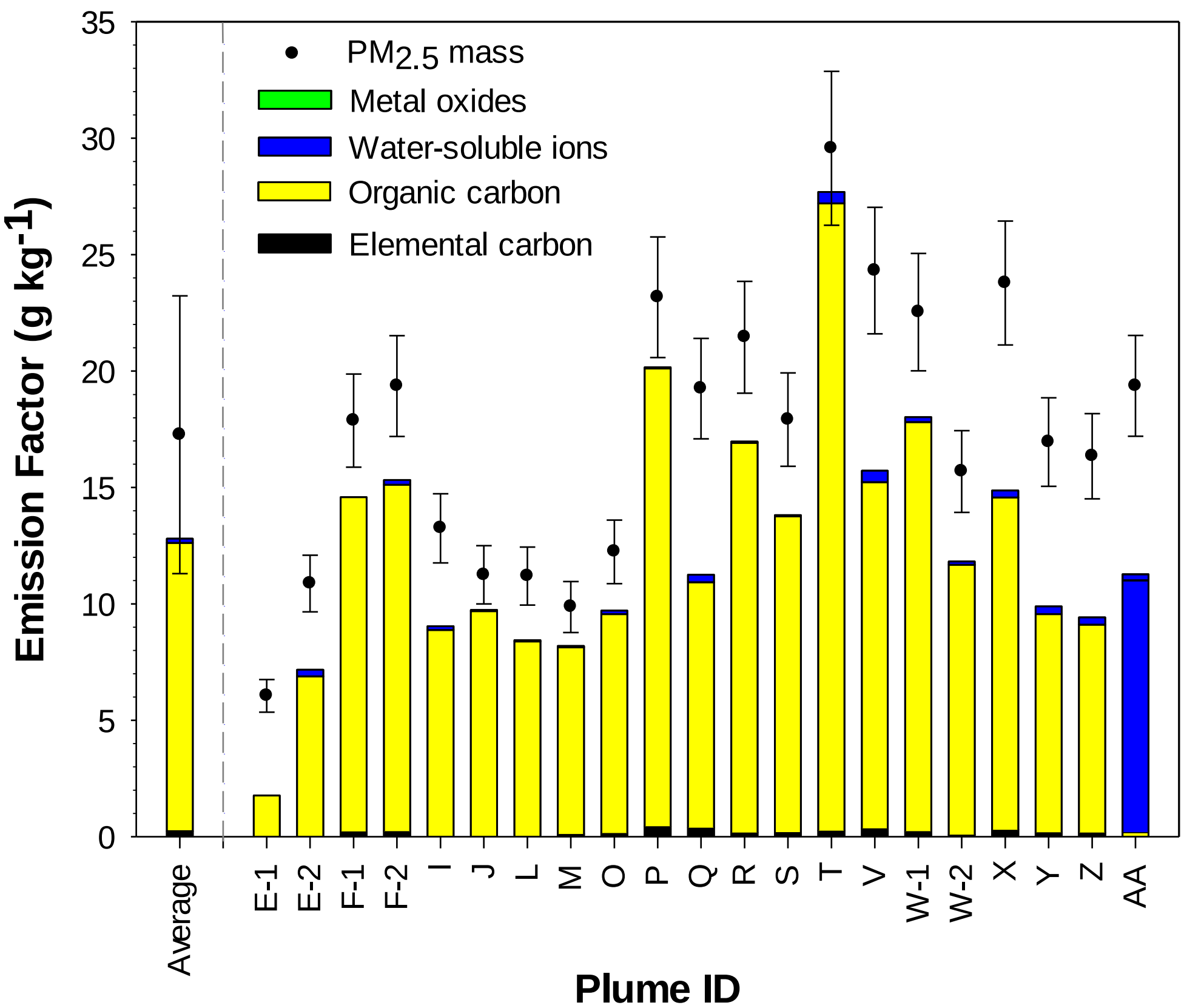

**Figure 2:** Linear regression of the measured organic carbon (OC) concentration versus the estimated organic matter (OM) concentration in sampled plumes that was calculated as the difference between $PM_{2.5}$ mass and the sum of EC, water-soluble ions and metal oxides.

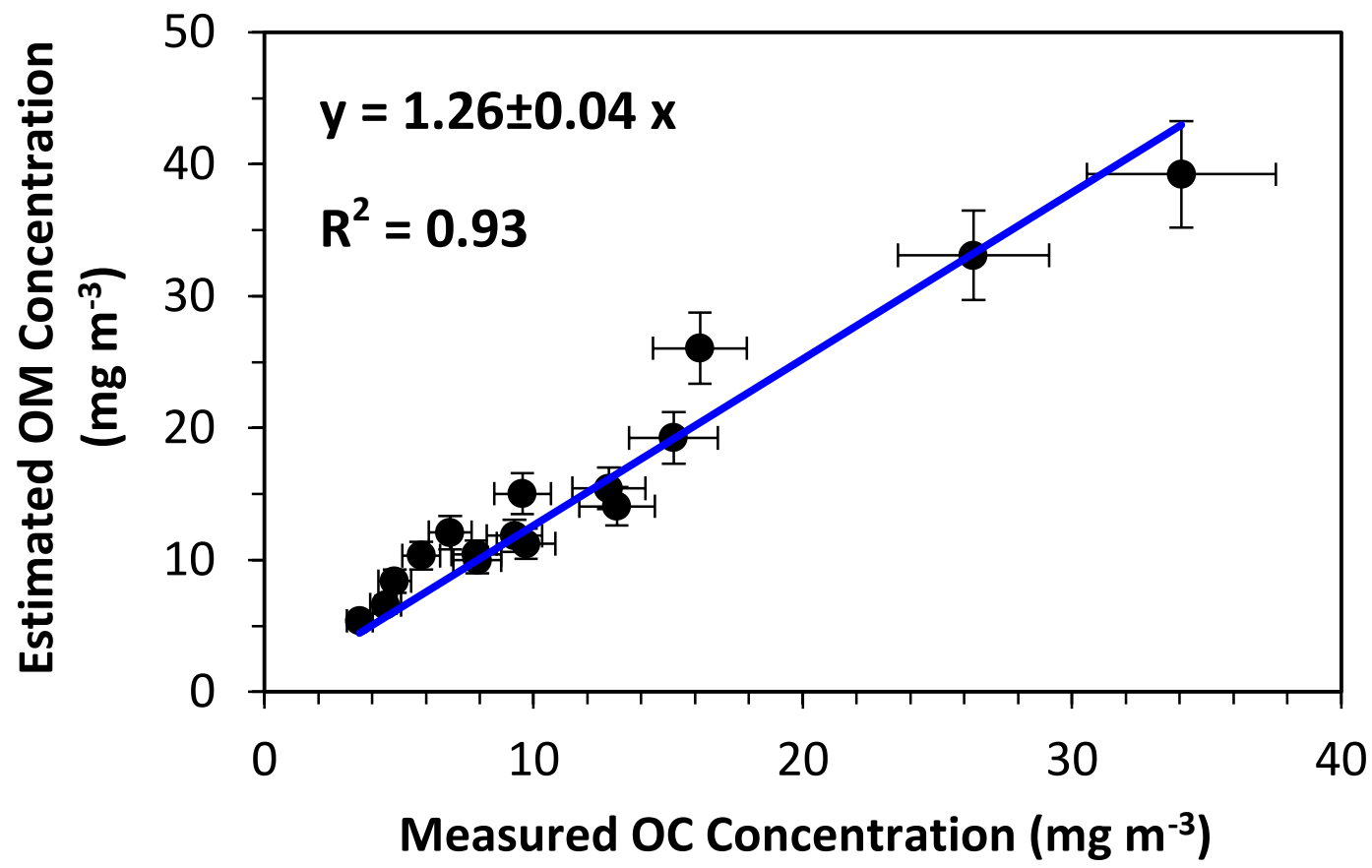

**Figure 3:** Organic carbon mass fraction of the speciated compound classes in selected peat burning emission samples. Plume Y was excluded from the average calculation as discussed in section 2.1.

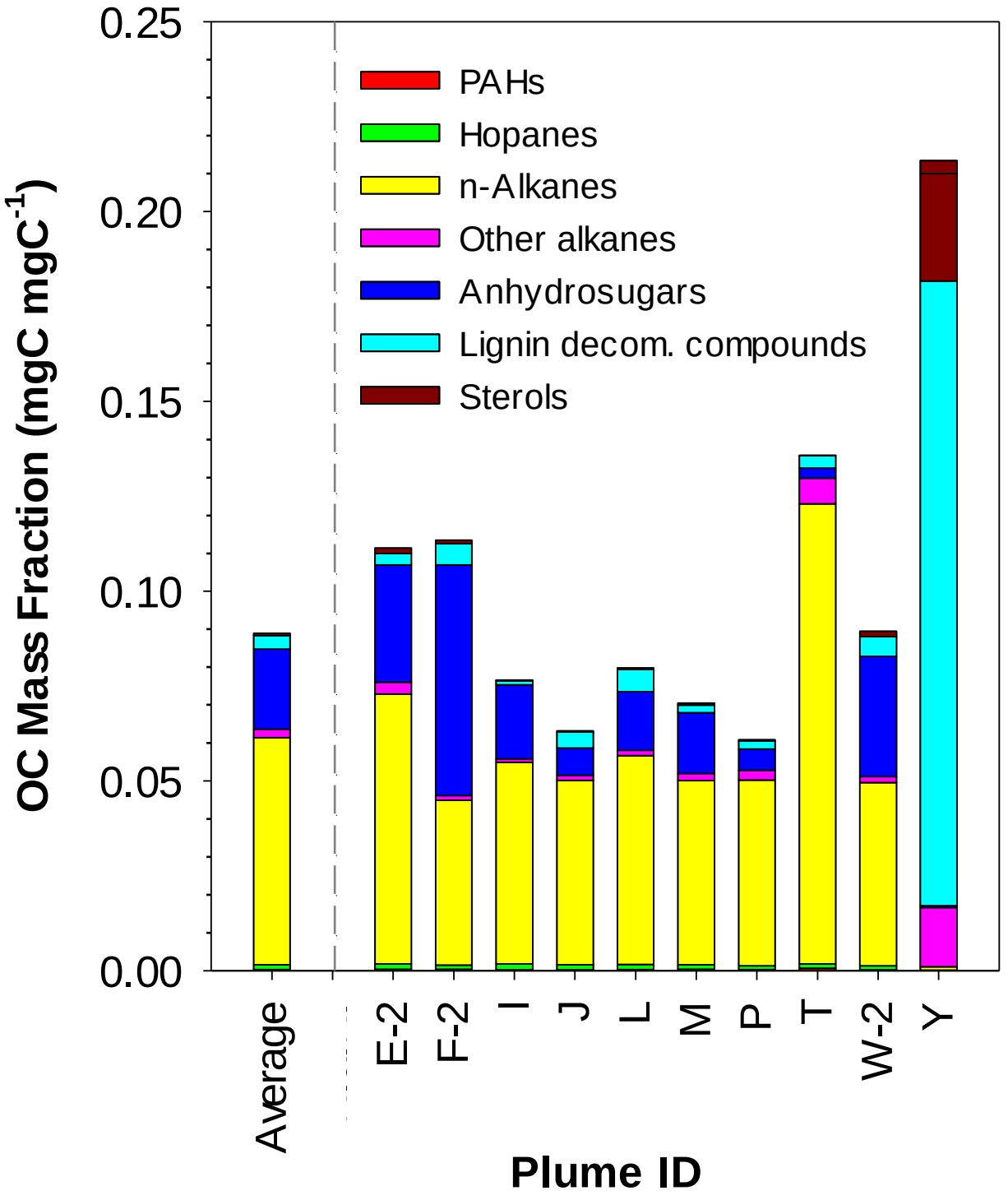

**Figure 4:** Molecular distribution of n-alkanes for selected plumes (n=10). Y axis indicates individual n-alkane mass fraction of OC. The horizontal lines (black) in the box represent the 25th, 50th (median) and 75th percentiles and mean values are indicated by the blue lines.

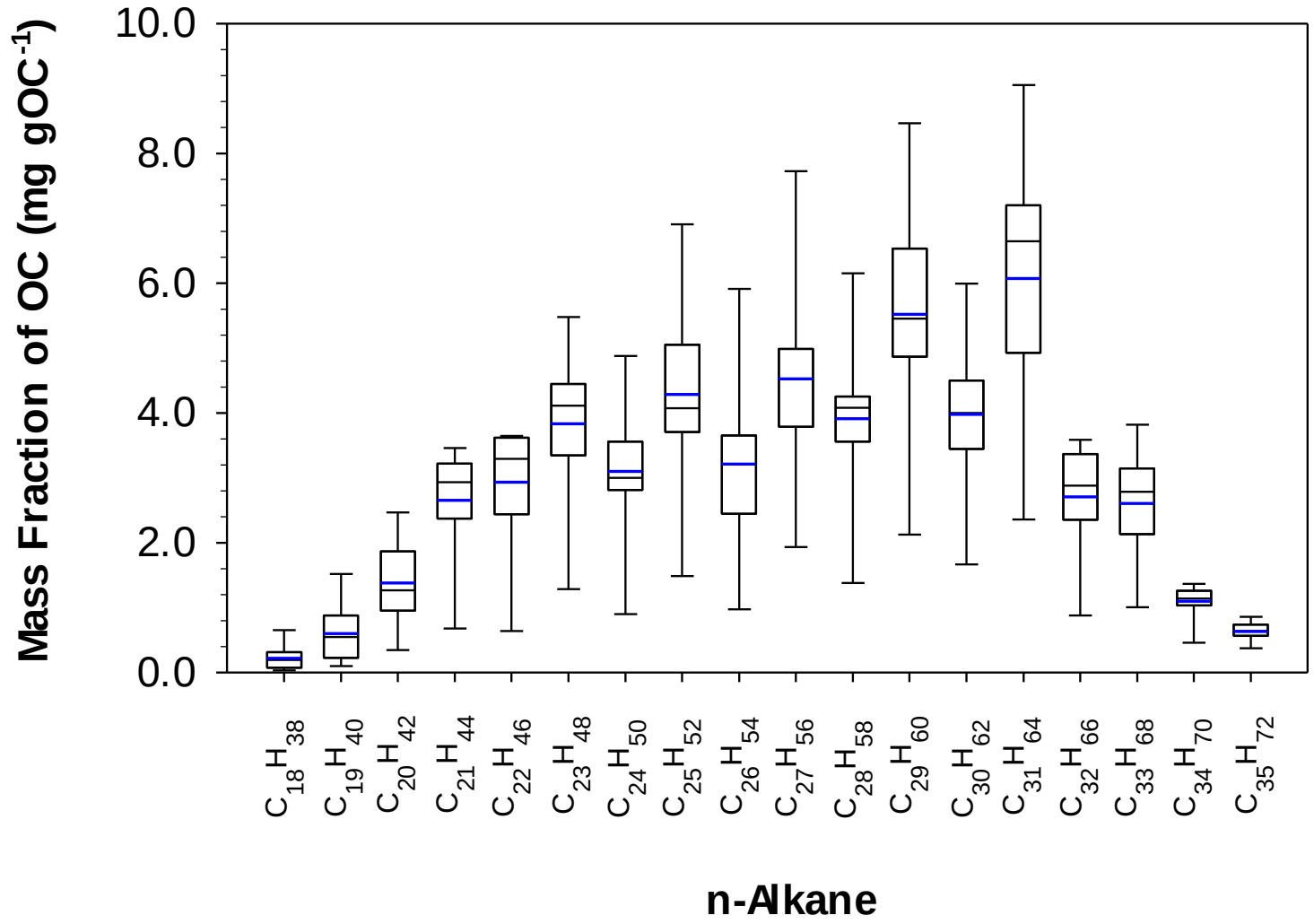

**Figure 5:** Organic carbon mass fractions of select anhydrosugars for study average and selected individual plumes. Plume Y was not included in average calculation as discussed in section 2.1. On average, the galactosan mass fraction was 0.14 mg $gOC^{-1}$ (maximum = 0.77 mg $gOC^{-1}$); due to its low concentrations, it was not included in the plot.

5 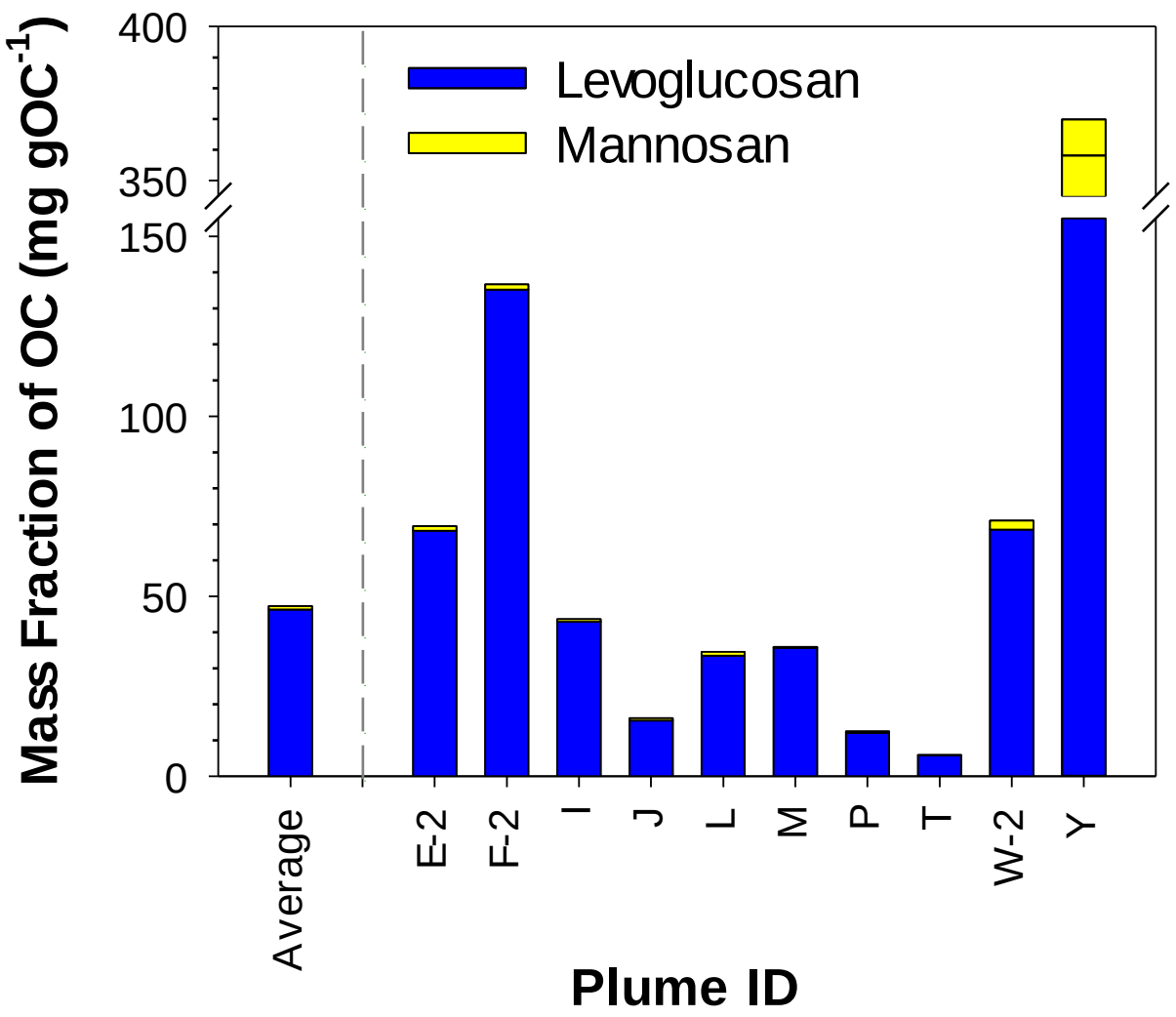

**Figure 6:** Organic carbon mass fraction of measured lignin decomposition products for study average and selected individual plumes. Plume Y was not included in average calculation as discussed in section 2.1.

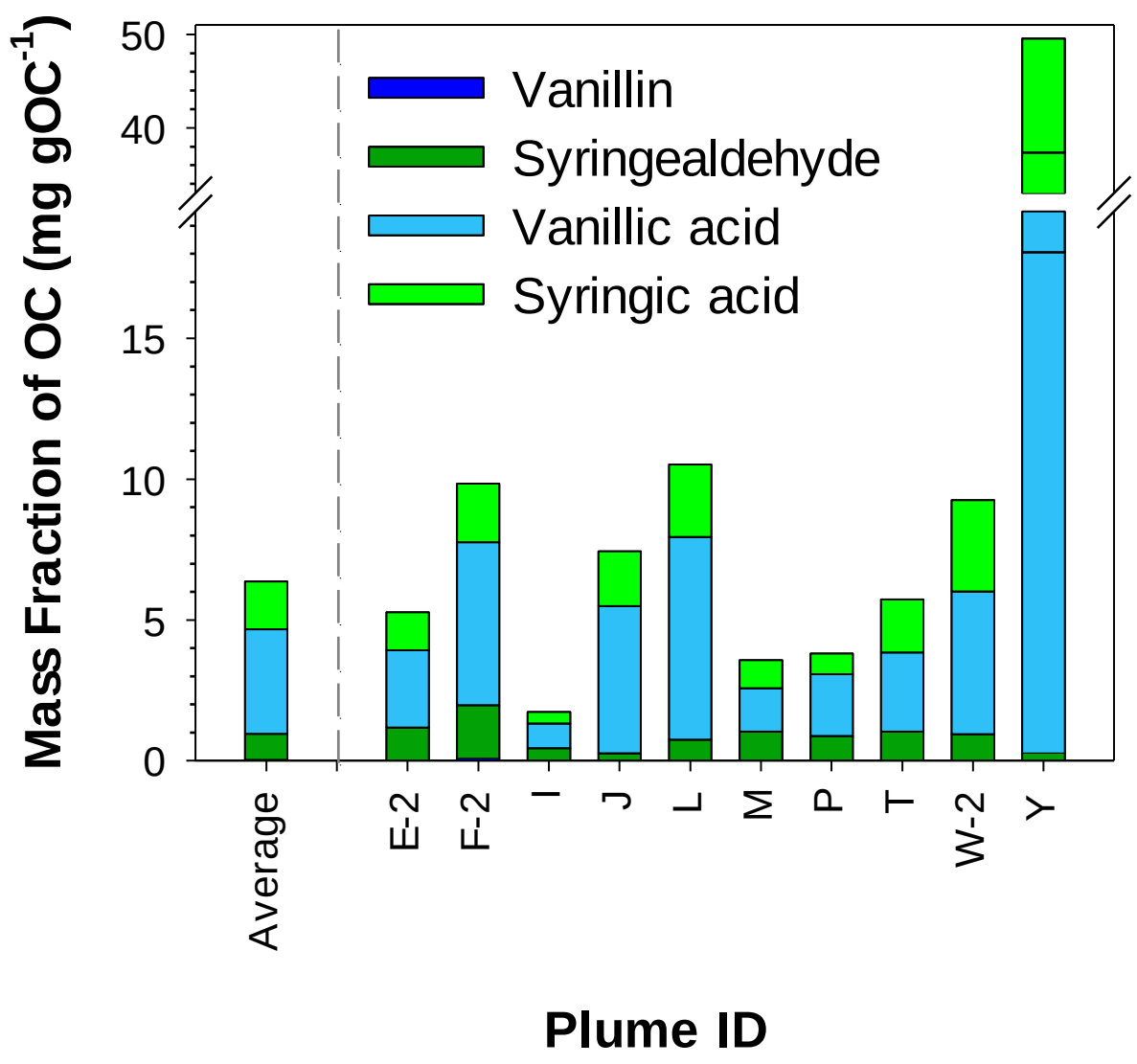

**Figure 7:** Emission ratios of vanillic acid to syringic acid.

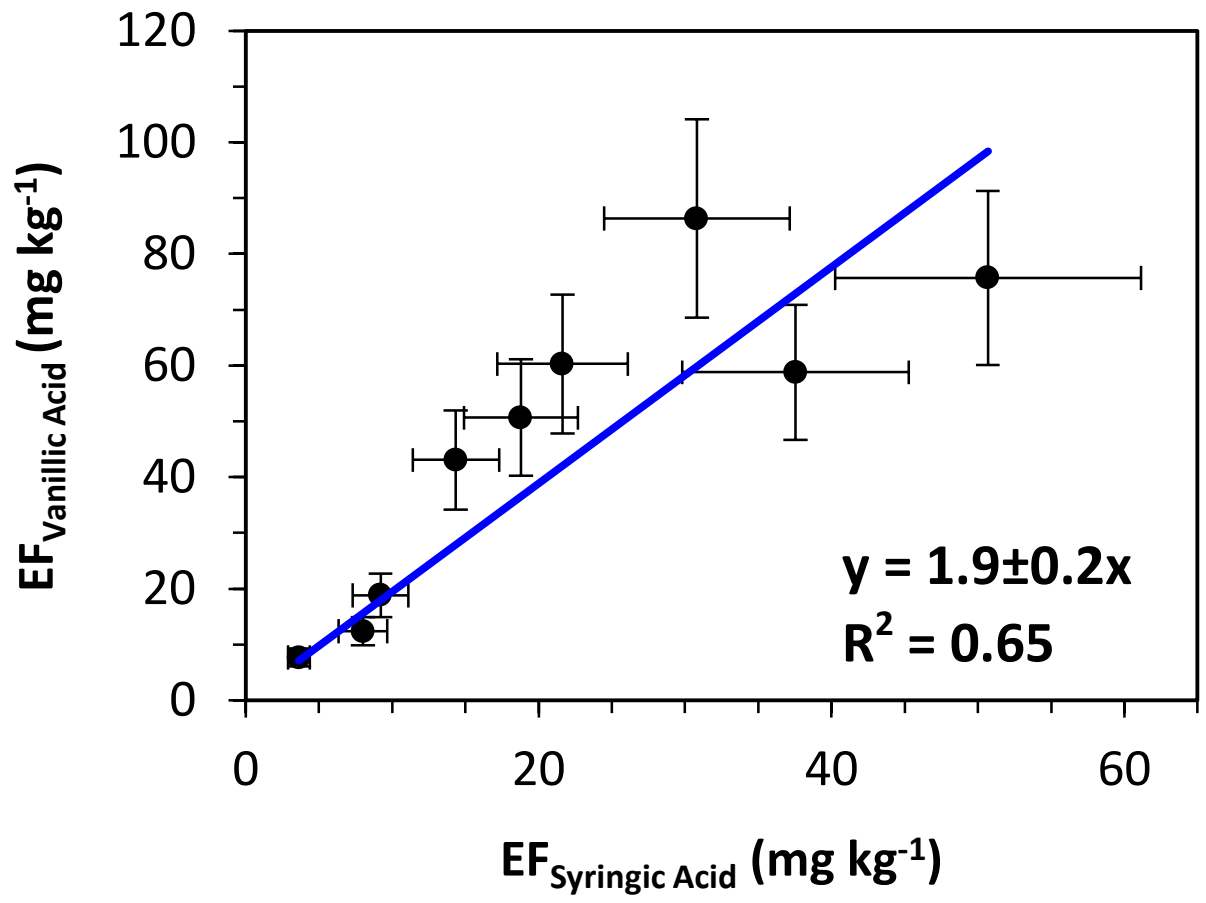

**Table 1:** Comparison of the averaged *in situ* Indonesian peat emission data to prior laboratory studies of peat combustion.

| **Peat location of origin (and type)** | **PM Size** | **No. of samples** | **$EF_{PM}$ ($g\ kg^{-1}$)** | **OC (% Mass)** | **EC (% Mass)** | **WSOC % of OC** | **OC:EC** | **MCE** | **Reference** |
|---|---|---|---|---|---|---|---|---|---|
| Indonesia | $PM_{2.5}$ | 21 | 17 | 72 | 1.1 | 16 | 60 | 0.78 | This study |
| Indonesia | $PM_{2.5}$ | 1 | 6.06[a] | 99[a] | 1[a] | - | 151 | 0.838 | Christian et al. (2003) |
| Indonesia | $PM_1$ | 1 | 34.9 | 99[b] | 0.03[c] | - | - | 0.891 | May et al. (2014) |
| Indonesia | $PM_{10}$ | 1 | 33 | 24 | 1.7 | 39 | 14 | - | Iinuma et al. (2007) |
| German | $PM_{10}$ | 1 | 44 | 29 | 2.2 | 52 | 13 | - | Iinuma et al. (2007) |
| North Carolina (ARNWR)[d] | $PM_{2.5}$ | 4 | 7.1 | 89 | 0.73 | - | 122 | 0.89 | Black et al. (2016) |
| North Carolina (PLNWR)[e] | $PM_{2.5}$ | 4 | 5.9 | 73 | 1.4 | - | 52 | 0.88 | Black et al. (2016) |
| North Carolina (ARNWR)[d] | $PM_{2.5}$ | 4 | 48-66 | - | - | - | - | 0.79-0.86 | Geron and Hays (2013) |
| North Carolina (PLNWR)[e] | $PM_{2.5}$ | 4 | 35-55 | - | - | - | - | 0.77-0.83 | Geron and Hays (2013) |
| North Carolina (Green Swamp) | $PM_{2.5}$ | 4 | 44-53 | - | - | - | - | 0.80-0.81 | Geron and Hays (2013) |
| Florida (sawgrass)[f] | $PM_{2.5}$ | 6 | 30 | - | - | - | - | - | McMahon et al. (1980) |
| Alaska (tundra core) | TSP | - | 41.3 | 93.5 | 2.6 | - | 36 | 0.87 | Chen et al. (2007) |

a) PM mass was not directly measured and was estimated as the sum of EC and OC; b) measured as organic aerosol; c) measured as refractory black carbon; d) Alligator River National Wildlife Refuge; e) Pocosin Lakes National Wildlife Refuge; f) corresponds to dry peat within the first 24 hours of combustion.

**Table 2:** Average emission factors for $PM_{2.5}$, EC, OC, water-soluble ions, metals (as mass fraction of $PM_{2.5}$), and organic species normalized to organic carbon mass. Individual EF data is given in Table S1.

| **Species** | **Study Average** | **Standard Deviation** |
|---|---|---|
| EF $PM_{2.5}$ (g $kg^{-1}$) | 17.3 | 6.0 |
| EC (as mass fraction of $PM_{2.5}$; g $gPM_{2.5}^{-1}$) | 0.011 | 0.005 |
| OC (as mass fraction of $PM_{2.5}$; g $gPM_{2.5}^{-1}$) | 0.72 | 0.11 |
| Water-soluble OC fraction | 0.16 | 0.11 |
| Water-insoluble OC fraction | 0.84 | 0.11 |
| Water-soluble ions (as mass fraction of $PM_{2.5}$; mg $gPM_{2.5}^{-1}$) | | |
| Sodium | 0.054 | 0.065 |
| Ammonium | 5.1 | 3.0 |
| Potassium | 0.26 | 0.43 |
| Fluoride | 0.66 | 0.63 |
| Chloride | 4.2 | 2.4 |
| Nitrate | 0.16 | 0.13 |
| Sulfate | 1.41 | 1.42 |
| Metals (as mass fraction of $PM_{2.5}$; µg $gPM_{2.5}^{-1}$) | | |
| Al | 0.113 | 0.059 |
| Ti | 0.083 | 0.056 |
| V | 0.048 | 0.021 |
| Mn | 0.058 | 0.031 |
| Ni | 0.019 | 0.011 |
| Sr | 0.059 | 0.030 |
| Ba | 0.40 | 0.19 |
| Organic species (as mass fraction of organic carbon; mg $gOC^{-1}$) | | |
| PAHs | | |
| Anthracene | 0.0062 | 0.0036 |
| Fluoranthene | 0.036 | 0.017 |
| Pyrene | 0.056 | 0.031 |
| Methylfluoranthene | 0.043 | 0.021 |
| Benzo(ghi)fluoranthene | 0.0056 | 0.0029 |
| Cyclopenta(cd)pyrene | 0.0045 | 0.0022 |
| Benz(a)anthracene | 0.023 | 0.013 |
| Chrysene | 0.054 | 0.021 |
| 1-Methylchrysene | 0.019 | 0.010 |
| Retene | 0.031 | 0.028 |

| | | |
|---|---|---|
| Benzo(b)fluoranthene | 0.023 | 0.013 |
| Benzo(k)fluoranthene | 0.0036 | 0.0028 |
| Benzo(j)fluoranthene | 0.0031 | 0.0023 |
| Benzo(e)pyrene | 0.029 | 0.016 |
| Benzo(a)pyrene | 0.0081 | 0.0066 |
| Perylene | 0.0041 | 0.0034 |
| Benzo(ghi)perylene | 0.016 | 0.011 |
| Dibenz(ah)anthracene | 0.0098 | 0.0085 |
| Picene | 0.0139 | 0.0051 |
| Hopanes | | |
| 17α(H)-22,29,30-Trisnorhopane | 0.344 | 0.058 |
| 17β(H)-21α (H)-30-Norhopane | 0.85 | 0.13 |
| 17α(H)-21β(H)-Hopane | 0.218 | 0.066 |
| *n*-Alkanes | | |
| Octadecane | 0.39 | 0.46 |
| Nonadecane | 1.1 | 1.3 |
| Eicosane | 2.2 | 2.2 |
| Heneicosane | 3.8 | 2.8 |
| Docosane | 4.3 | 3.2 |
| Tricosane | 4.8 | 2.1 |
| Tetracosane | 4.1 | 2.2 |
| Pentacosane | 5.4 | 2.4 |
| Hexacosane | 4.1 | 2.1 |
| Heptacosane | 5.5 | 2.2 |
| Octacosane | 4.8 | 2.0 |
| Nonacosane | 6.5 | 1.9 |
| Triacontane | 4.7 | 1.4 |
| Hentriacontane | 6.7 | 1.4 |
| Dotriacontane | 3.03 | 0.52 |
| Tritriacontane | 2.83 | 0.54 |
| Tetratriacontane | 1.25 | 0.23 |
| Pentatriacontane | 0.66 | 0.15 |
| Heptatriacontane | 0.82 | 0.26 |
| Octriacontane | 2.5 | 1.3 |
| Nonatriacontane | 0.98 | 0.47 |
| Branched Alkanes | | |
| Norpristane | 0.35 | 0.47 |
| Pristane | 1.0 | 1.2 |

| | | |
|---|---|---|
| Squalane | 1.31 | 0.74 |
| Anhydrosugars | | |
| Levoglucosan | 46 | 40 |
| Mannosan | 0.93 | 0.76 |
| Galactosan | 0.14 | 0.13 |
| Lignin Decomposition Products | | |
| Vanillin | 0.030 | 0.044 |
| Syringealdehyde | 0.93 | 0.46 |
| Vanillic acid | 3.7 | 2.2 |
| Syringic acid | 1.69 | 0.91 |
| Sterols | | |
| Stigmasterol | 0.22 | 0.11 |
| β-Sitosterol | 0.53 | 0.34 |
| Campesterol | 0.29 | 0.20 |

**Table 3:** Estimated emissions from Indonesian peat fires during 2015 El Niño, based on a burned area of $8.5 \times 10^5$ ha (Whitburn et al., 2016), an average burning depth of 34±12 cm (Stockwell et al, 2016), and peat bulk density 0.120±0.005 g $cm^{-3}$ (Konecny et al., 2016). The uncertainty of the estimated value is propagated using standard deviations of the mean EFs, burn depth and peat bulk density.

| **Species** | **Total Estimated Emission** | |
|---|---|---|
| | **C-mass based (Tg C)** | **Mass based (Tg)** |
| $PM_{2.5}$ | - | 6.0±5.5 |
| C-containing compounds | | |
| $OC_{(PM2.5)}$ | 4.3±4.3 | - |
| $EC_{(PM2.5)}$ | 0.083±0.081 | - |
| $CO_{2(g)}$[a] | 149±71 | 547±259 |
| $CO_{(g)}$[a] | 44±30 | 102±69 |
| $CH_{4(g)}$[a] | 2.5±2.6 | 3.3±3.5 |
| Other C-containing trace gases[a] | 5.5±1.3 | 9.3±2.6 |
| Total C | 205±77 | - |
| Water-soluble ions in $PM_{2.5}$ | | |
| $NH_4^+$ | - | 0.032±0.039 |
| $Cl^-$ | - | 0.026±0.032 |
| $NO_3^-$ | - | 0.0010±0.0013 |
| $SO_4^{2-}$ | - | 0.0096±0.0151 |
| Other atmospheric gases | | |
| $NH_{3(g)}$[a] | - | 1.00±0.91 |
| $HCl_{(g)}$[a] | - | 0.012±0.014 |
| $NO_{(g)}$[a] | - | 0.11±0.17 |
| $HONO_{(g)}$[a] | - | 0.073±0.061 |

a-EFs are based on Stockwell et al., (2016)

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
