# Peer review of "Chemical characterization of fine particulate matter emitted by peat fires in Central Kalimantan, Indonesia, during the 2015 El Niño"

_Atmospheric Chemistry and Physics, 2017_

## Referee Comment (RC1) · Anonymous Referee #2 · 8 Oct 2017

1. Given there are only few measurements of smoke aerosol properties in Maritime Continents (MC), the work presented here surely is timely and highly relevant to the ACP. The paper is missing a body of literature on the work recently done by 7SEAS program in this region, particularly those paper in the 7SEAS' special issue published in Atmospheric Research in 2013. Note, the link between El Nino and fires still remain unclear; what is known in this part of the world is that Hurricane in subtropical Philippines can lead to large fires in Indonesia. This is worth mentioning, as the paper seems to say that it is the El Nino year that made this study interesting - which is not.

[Figure]

Reid, J., E. Hyer, R. Johnson, B. N. Holben, J. Zhang, J. R. Campbell, S. A. Christopher, L. D. Girolamo, L. Giglio, R. E. Holz, C. Kearney, J. Miettinen, E. A. Reid, F. J. Turk, J. Wang, P. Xian, R. J. Yokelson, G. Zhao, R. Balasubramanian, B.-N. Chew, S. Janai, N. Lagrosas, P. Lestari, N.-H. Lin, M. Mahmud, B. Norris, A. X. Nguyen, N. T. K. Oahn, M. Oo, S. Salinas, and S.-C. Liew, 2013. Observing and understanding the Southeast Asia aerosol system by remote sensing: An initial review and analysis for the Seven Southeast Asian Studies (7SEAS) program, Atmospheric Research , 122, 403-468.

Wang, J., C. Ge, Z. Yang, E. J. Hyer, J. S. Reid, B.-N. Chew, M. Mahmud, Y. Zhang, and M. Zhang, 2013. Mesoscale modeling of smoke transport over the Southeast Asian Maritime Continent: interplay of sea breeze, trade wind, typhoon, and topography, Atmospheric Research , 122, 486-503.

2. Relevant work regarding the importance of smoke aerosol composition on regional climate can be found below. It is important to discuss if the past modeling work in this region, based on your data of smoke optical properties, is good enough or has large uncertainties - likely a huge overestimation or underestimation of smoke absorption? Such discussion is important as the abstract of this manuscript says so, yet the manuscript itself touched very little on the recent modeling work of smoke radiative effects in that region.

Ge, C., J. Wang , and J. S. Reid, 2014, Mesoscale modeling of smoke transport over the Southeast Asian Maritime Continent: coupling of smoke direct radiative feedbacks below and above the low-level clouds, Atmos. Chem. Phys. , 14, 159-174.

3. Finally, the fire emission inventory still has large uncertainty, and it is unclear how the measured results here compare with the results widely used by different inventories. Can we say OC/BC ratio uncertainty or variation is a factor of 2 or 3? See below several papers and references therein.

Zhang, F., J. Wang , C. Ichoku, E. Hyer, Z. Yang, C. Ge, S. Su, X. Zhang, S. Kondragunta, J. Kaiser, C. Wiedinmyer, and A. da Silva, 2014. Sensitivity of mesoscale

modeling of smoke direct radiative effect to the emission inventory: A case study in northern sub-Saharan African region, Environmental Research Letter, 9, 075002.

Koppmann, R., K Czapiewski, JS Reid, A review of biomass burning emissions, part I: gaseous emissions of carbon monoxide, methane, volatile organic compounds, and nitrogen containing compounds R Koppmann, K Czapiewski, JS Reid - Atmospheric Chemistry and Physics, 2005.

Reid, J. S., Koppmann, R., Eck, T. F., and Eleuterio, D. P.: A review of biomass burning emissions part II: intensive physical properties of biomass burning particles, Atmos. Chem. Phys., 5, 799-825, 2005.

My recommendation is that the importance of BC/OC ratio measured in this paper should be discussed in the context of these past work, so that, as said in the abstract, these measurments are valuable for the emission inventory community and atmospheric modeling community.

---

## Referee Comment (RC2) · Anonymous Referee #1 · 11 Oct 2017

Jayarathne et al. characterized in-situ particulate matter emitted from 18 peatland fire plumes in Indonesia. The authors have performed thorough and careful analysis of their samples, including an impressive suite of organic and inorganic chemical analyses. They determined that PM emissions from peat fires are overwhelmingly composed of organic carbon that is largely hydrophobic and with a lower OM:OC than observed in other biomass burning experiments. The paper is well written and will be of interest to the scientific community. I recommend publication following the minor corrections and clarifications noted below.

[Figure]

Specific comments: Page 3, line 2: "Thus, a mobile lab. . ." The end of this paragraph feels out of place and would fit better merged with the last paragraph of the Introduction.

Page 4, line 3: Missing space in "spreadslowly".

Page 4, line 8: 'which' should be 'with'.

Page 7, line 11: "The samples were collected directly from visible plumes in smoldering peat." Approximately how far from the smoldering peat were the samples collected? This is relevant later in the text when comparing measured EFs to laboratory studies of peat fires (e.g., pg 15, line 7).

Page 8, line 4: "the plume of smoke cooled to near-ambient temperature, to allow for gas-particle partitioning to equilibrate prior to sample collection." Gas-particle partitioning will continue to change at ambient temperature due to plume dilution. Please rephrase the sentence.

Pg 14, lines 8-10: "The percent difference across duplicate samples was 57%, 37%, and 8% for plumes E, F, and W, respectively, indicating temporal variability in emissions from a single plume as the peat fire progresses." Please add further details regarding the timing of the duplicate samples. "Duplicate" implies parallel sampling, whereas the quoted discussion suggests sequential sampling.

Pg. 20, line 17: Missing space "emissionsfrom"

Pg 21, lines 13-17: Has the VA:SA ratio been measured in smoke from other fuel types? Is a ratio of ∼1.9 specific to peat smoke or biomass burning smoke in general?

Pg. 27, line 12: typo "peatl"

---

## Author Comment (AC1) · 15 Dec 2017

Referee #1 General Comments: Jayarathne et al. characterized in-situ particulate matter emitted from 18 peatland fire plumes in Indonesia. The authors have performed thorough and careful analysis of their samples, including an impressive suite of organic and inorganic chemical analyses. They determined that PM emissions from peat fires are overwhelmingly composed of organic carbon that is largely hydrophobic and with a lower OM:OC than observed in other biomass burning experiments. The paper is well written and will be of interest to the scientific community. I recommend publication

following the minor corrections and clarifications noted below.

Response to Referee #1 General Comments: We thank the reviewer for their assessment of the manuscript and their suggestions to improve it. We have incorporated their suggestions into the revised manuscript and detail the changes in response to their specific comments below.

Referee #1 Specific Comment 1: Page 3, line 2: "Thus, a mobile lab. . ." The end of this paragraph feels out of place and would fit better merged with the last paragraph of the Introduction.

Response to Referee #1 Specific Comment 1: As suggested by the reviewer, we have moved the text previously located at the end of the first paragraph of the introduction to the beginning of the last paragraph of the introduction.

Referee #1 Specific Comment 2: Page 4, line 3: Missing space in "spreadslowly".

Response to Referee #1 Specific Comment 2: We thank the reviewer for pointing this out and have corrected this as suggested.

Referee #1 Specific Comment 3: Page 4, line 8: 'which' should be 'with'.

Response to Referee #1 Specific Comment 3: We agree with the reviewer and have revised the text as suggested.

Referee #1 Specific Comment 4: Page 7, line 11: "The samples were collected directly from visible plumes in smoldering peat." Approximately how far from the smoldering peat were the samples collected? This is relevant later in the text when comparing measured EFs to laboratory studies of peat fires (e.g., pg 15, line 7).

Response to Referee #1 Specific Comment 4: We agree that this information is important to include and have changed the text at the beginning of section 2.2 on Sample Collection to now read: "The sampling inlet was mounted on a ~2.5 m pole to allow sampling of smoke from a safe distance. The inlet was positioned approximately 2-3 m

downwind of the smoldering peat at a point where the plume of smoke had cooled to near-ambient temperature."

Referee #1 Specific Comment 5: Page 8, line 4: "the plume of smoke cooled to near-ambient temperature, to allow for gas-particle partitioning to equilibrate prior to sample collection." Gas-particle partitioning will continue to change at ambient temperature due to plume dilution. Please rephrase the sentence.

Response to Referee #1 Specific Comment 5: We agree with the reviewer and have removed the phrase implying complete equilibration as shown just above. The aerosol evolution over its complete lifetime is beyond the scope of this paper.

Referee #1 Specific Comment 6: Pg 14, lines 8-10: "The percent difference across duplicate samples was 57%, 37%, and 8% for plumes E, F, and W, respectively, indicating temporal variability in emissions from a single plume as the peat fire progresses." Please add further details regarding the timing of the duplicate samples. "Duplicate" implies parallel sampling, whereas the quoted discussion suggests sequential sampling.

Response to Referee #1 Specific Comment 6: We agree with the reviewer that the collected samples are not duplicates and indeed were collected in sequence. We have made several improvements to clarify this:

In section 2.1 we now state: "Two PM samples were collected from plumes E, F and W, bringing the total number of PM samples to 21. Because of the variability in PM emissions within a single plume, both values were used in calculating study-averages.

In section 2.2 we now state: "PM samples were collected over a period of 9-30 minutes each, at PM2.5 concentrations that averaged 15 mg m-3 and ranging from 1-40 mg m-3. The duration of filter sample collection and PM2.5 concentrations sampled are summarized in Table S1 for each plume. For plumes with two samples collected, the time over which samples were collected were comparable and the sampled PM2.5

concentrations were within a factor of three."

In the footnote to Table S1, we added a reference to Stockwell et al. (2016, Table S1), which provides additional details including sampling location, peat type, burning inclusions, burn depth, surface fuels, temperature, relative humidity, wind, and other sampling notes.

And finally, section 3.1 now reads: "The percent difference across samples collected sequentially from the same plume was 57%, 37%, and 8% for plumes E, F, and W, respectively, indicating temporal variability in emissions within the fire as it progresses."

Referee #1 Specific Comment 7: Pg. 20, line 17: Missing space "emissionsfrom"

Response to Referee #1 Specific Comment 7: We thank the reviewer for pointing out this typo and have revised the text as suggested.

Referee #1 Specific Comment 8: Pg 21, lines 13-17: Has the VA:SA ratio been measured in smoke from other fuel types? Is a ratio of _1.9 specific to peat smoke or biomass burning smoke in general?

Response to Referee #1 Specific Comment 8: We agree with the reviewer that it is necessary to further elaborate upon this point. We have revised text in section 3.5.3 to read:

"3.5.3 Lignin decomposition compounds Syringaldehyde (S), vanillin (V), syringic acid (SA) and vanillic acid (VA) derived from lignin pyrolysis were quantified, with a combined EF ranging 15-154 mg kg-1 and averaging 80±50 mg kg-1 (Table S1). Correlations among aldehydes (V and S) were not significant, possibly due to V partitioning to the gas phase, as indicated by its detection on backup filters, whereas other species (S, VA, and SA) were detected only on front filters indicative of particle phase species. We examined the potential of the VA:SA ratios to be useful in distinguishing this source from other types of biomass burning, since VA:SA depends on the lignin composition of the biomass (Simoneit et al., 1999). A significant moderate correlation was observed

between EFVA and EFSA (R2=0.65; p=0.004). Based on linear regression analysis, VA:SA was found to be 1.9±0.2 for freshly emitted peat smoke in this study (Figure 7). This value agrees well with observations of VA:SA in PM2.5 in Malaysia affected by Sumatran peat fires, which had a VA:SA ratio of 1.7±0.4 (Fujii et al., 2015b) and the ratio of vanillyl phenols to syringyl phenols ratio of 2.0 reported for Kalimantan peat (Orem et al., 1996). Meanwhile, other studies indicate lower VA:SA ratios for near-source emissions of Sumatran peat burning (1.1±0.4) (Fujii et al., 2015a) and laboratory burning of South Sumatran peat (0.11) (Iinuma et al., 2007). Because other biomasses in South Asia have VA:SA that fall in this range, such as bamboo (1.17) and sugar cane (1.78) (Simoneit et al., 1999), this ratio is unlikely to be useful in distinguishing peat burning from other types of biomass burning in the absence of other distinguishing chemical or physical properties. Further, syringyl compounds degrade more quickly in peat compared to vanillyl compounds (Orem et al., 1996) and post-emission SA degrades more quickly than VA by photolysis in the atmosphere, such that VA:SA is likely to increase with smoke transport (Fujii et al., 2015b). Consequently, this ratio has limited utility in source identification and apportionment."

Referee #1 Specific Comment 9: Pg. 27, line 12: typo "peatl"

Response to Referee #1 Specific Comment 9: We thank the reviewer for pointing out this typo and have revised the text as suggested.

Works Cited

Fujii, Y., Kawamoto, H., Tohno, S., Oda, M., Iriana, W., and Lestari, P.: Characteristics of carbonaceous aerosols emitted from peatland fire in Riau, Sumatra, Indonesia (2): Identification of organic compounds, Atmospheric Environment, 110, 1-7, 10.1016/j.atmosenv.2015.03.042, 2015a.

Fujii, Y., Tohno, S., Amil, N., Latif, M. T., Oda, M., Matsumoto, J., and Mizohata, A.: Annual variations of carbonaceous PM2.5 in Malaysia: influence by Indonesian peatland fires, Atmospheric Chemistry and Physics, 15, 13319-13329, 10.5194/acp-15-13319-

2015, 2015b.

Iinuma, Y., Bruggemann, E., Gnauk, T., Muller, K., Andreae, M. O., Helas, G., Parmar, R., and Herrmann, H.: Source characterization of biomass burning particles: The combustion of selected European conifers, African hardwood, savanna grass, and German and Indonesian peat, Journal of Geophysical Research-Atmospheres, 112, 26, D08209, 10.1029/2006jd007120, 2007.

Orem, W. H., Neuzil, S. G., Lerch, H. E., and Cecil, C. B.: Experimental early-stage coalification of a peat sample and a peatified wood sample from Indonesia, Organic Geochemistry, 24, 111-125, https://doi.org/10.1016/0146-6380(96)00012-5, 1996.

Simoneit, B. R., Schauer, J. J., Nolte, C., Oros, D. R., Elias, V. O., Fraser, M., Rogge, W., and Cass, G. R.: Levoglucosan, a tracer for cellulose in biomass burning and atmospheric particles, Atmospheric Environment, 33, 173-182, 10.1016/S1352-2310(98)00145-9, 1999.

Please also note the supplement to this comment:
https://www.atmos-chem-phys-discuss.net/acp-2017-608/acp-2017-608-AC1-supplement.pdf

**Supplement:**

**Table S1:** Summary of filter samples analyzed in this study, by plume, duration of filter sample collection, and PM$_{2.5}$ mass concentrations in sampled smoke plumes (with uncertainty).

| Plume[1] | Duration of sampling (min:sec) | PM mass (mg m$^{-3}$) | PM mass uncertainty (mg m$^{-3}$) |
|---|---|---|---|
| A-2 | 20:09 | 1.96E+01 | 1.96E+00 |
| B | 16:02 | 1.30E+00 | 1.30E-01 |
| E-1 | 20:04 | 1.81E+00 | 1.81E-01 |
| E-2 | 19:10 | 5.60E+00 | 5.60E-01 |
| F-1 | 21:09 | 1.77E+01 | 1.77E+00 |
| F-2 | 19:31 | 1.97E+01 | 1.97E+00 |
| I | 21:28 | 6.73E+00 | 6.73E-01 |
| J | 20:05 | 1.13E+01 | 1.13E+00 |
| L | 20:04 | 1.05E+01 | 1.05E+00 |
| M | 20:08 | 1.58E+01 | 1.58E+00 |
| O | 19:56 | 1.94E+01 | 1.94E+00 |
| P | 11:25 | 4.02E+01 | 4.02E+00 |
| Q | 30:39 | 1.26E+01 | 1.26E+00 |
| R | 31:53 | 1.01E+01 | 1.01E+00 |
| S | 30:29 | 1.20E+01 | 1.20E+00 |
| T | 29:53 | 1.45E+01 | 1.45E+00 |
| V | 29:46 | 1.56E+01 | 1.56E+00 |
| W-1 | 09:20 | 3.39E+01 | 3.39E+00 |
| W-2 | 15:03 | 1.92E+01 | 1.92E+00 |
| X | 15:29 | 2.68E+01 | 2.68E+00 |
| Y | 20:05 | 1.39E+01 | 1.39E+00 |
| Z | 20:26 | 1.07E+01 | 1.07E+00 |
| AA | 20:00 | 8.66E+00 | 8.66E-01 |

1) Additional information each plume is available in Stockwell et al. (2016, Table S1), including sampling location, peat type, burning inclusions, burn depth, surface fuels, temperature, relative humidity, wind, and other sampling notes.

---

## Author Comment (AC2) · 15 Dec 2017

Referee #2 Comment 1: Given there are only few measurements of smoke aerosol properties in Maritime Continents (MC), the work presented here surely is timely and highly relevant to the ACP. The paper is missing a body of literature on the work recently done by 7SEAS program in this region, particularly those paper in the 7SEAS' special issue published in Atmospheric Research in 2013. Note, the link between El Nino and fires still remain unclear; what is known in this part of the world is that Hurricane in subtropical Philippines can lead to large fires in Indonesia. This is worth mentioning,

as the paper seems to say that it is the El Nino year that made this study interesting - which is not. Reid, J., E. Hyer, R. Johnson, B. N. Holben, J. Zhang, J. R. Campbell, S. A. Christopher, L. D. Girolamo, L. Giglio, R. E. Holz, C. Kearney, J. Miettinen, E. A. Reid, F. J. Turk, J. Wang, P. Xian, R. J. Yokelson, G. Zhao, R. Balasubramanian, B.-N. Chew, S. Janai, N. Lagrosas, P. Lestari, N.-H. Lin, M. Mahmud, B. Norris, A. X. Nguyen, N. T. K. Oahn, M. Oo, S. Salinas, and S.-C. Liew, 2013. Observing and understanding the Southeast Asia aerosol system by remote sensing: An initial review and analysis for the Seven Southeast Asian Studies (7SEAS) program, Atmospheric Research, 122, 403-468.

Wang, J., C. Ge, Z. Yang, E. J. Hyer, J. S. Reid, B.-N. Chew, M. Mahmud, Y. Zhang, and M. Zhang, 2013. Mesoscale modeling of smoke transport over the Southeast Asian Maritime Continent: interplay of sea breeze, trade wind, typhoon, and topography, Atmospheric Research, 122, 486-503.

Response to Referee #2 Comment 1: We thank the reviewer for their review of this manuscript and their suggestions to improve it. The recommended references make an important point. While it is clear that smoke impacts peak in El-Nino years, the actual interannual variability in the amount of burning is harder to measure and multiple factors may influence that. As suggested, we added the recommended references to the introduction and have clarified the conditions that lead to large peat burning impacts in Indonesia. We have revised the first paragraph of the introduction with the following text:

"Major peat burning impacts have coincided with the El Niño Southern Oscillation (e.g., 1997-98, 2006, 2015), during which warmer conditions decrease dry season precipitation, which lowers the water table of peatlands, increases their flammability, and promotes longer-range transport of the smoke (Reid et al., 2013). Within a season, meteorological factors contribute to peat-burning pollution events and transport, including typhoons and wind patterns (Wang et al., 2013). Notably, even in non-El Niño years, peat burning remains an important source of biomass burning emissions in Southeast

Asia (Reid et al., 2013).The 2015 peatland fire episode that occurred September – November 2015 occurred during an El Niño year and was reported as the strongest peatland fire episode since 1997-98 (Parker et al., 2016; Koplitz et al., 2016; Huijnen et al., 2016). "

Referee #2 Comment 2: Relevant work regarding the importance of smoke aerosol composition on regional climate can be found below. It is important to discuss if the past modeling work in this region, based on your data of smoke optical properties, is good enough or has large uncertainties - likely a huge overestimation or underestimation of smoke absorption? Such discussion is important as the abstract of this manuscript says so, yet the manuscript itself touched very little on the recent modeling work of smoke radiative effects in that region.

Ge, C., J. Wang , and J. S. Reid, 2014, Mesoscale modeling of smoke transport over the Southeast Asian Maritime Continent: coupling of smoke direct radiative feedbacks below and above the low-level clouds, Atmos. Chem. Phys. , 14, 159-174.

Response to Referee #2 Comment 2: Modeling smoke climate impacts in SE Asia involves the initial emissions, transport, and evolution of smoke aerosol from multiple fuel types. The focus of this paper is the first in-situ PM measurements of the tropical peat fire emissions and we prefer not to broaden it to a comprehensive discussion of past modeling in the region. However, we agree with the reviewer that additional context is useful and we have added further discussion of the observed OC:EC ratios and their variability. The following text has been added as the third paragraph in section 3.2:

"The prior lack of information on light absorption by peat burning emissions could potentially limit the accuracy of direct radiative forcing estimates in Southeast Asia (Ge et al., 2014). Previously, Ge et al. (2014) modeled radiative forcing using OC:EC values up to 17. Our much larger OC:EC values could imply a more strongly scattering aerosol is relevant depending on the extent to which regional emissions are dominated

by peat burning. in addition, with new measurements of BrC presented in our companion paper (Stockwell et al., 2016), the role of BrC in direct radiative forcing should be evaluated in future assessments of this kind."

Referee #2 Comment 3: Finally, the fire emission inventory still has large uncertainty, and it is unclear how the measured results here compare with the results widely used by different inventories. Can we say OC/BC ratio uncertainty or variation is a factor of 2 or 3? See below several papers and references therein.

Zhang, F., J. Wang , C. Ichoku, E. Hyer, Z. Yang, C. Ge, S. Su, X. Zhang, S. Kondragunta, J. Kaiser, C. Wiedinmyer, and A. da Silva, 2014. Sensitivity of mesoscale modeling of smoke direct radiative effect to the emission inventory: A case study in northern sub-Saharan African region, Environmental Research Letter, 9, 075002.

Koppmann, R., K Czapiewski, JS Reid, A review of biomass burning emissions, part I: gaseous emissions of carbon monoxide, methane, volatile organic compounds, and nitrogen containing compounds R Koppmann, K Czapiewski, JS Reid – Atmospheric Chemistry and Physics, 2005.

Reid, J. S., Koppmann, R., Eck, T. F., and Eleuterio, D. P.: A review of biomass burning emissions part II: intensive physical properties of biomass burning particles, Atmos. Chem. Phys., 5, 799-825, 2005.

My recommendation is that the importance of BC/OC ratio measured in this paper should be discussed in the context of these past work, so that, as said in the abstract, these measurments are valuable for the emission inventory community and atmospheric modeling community.

Response to Referee #2 Comment 3: As stated above, models must consider multiple fuels, transport, and evolution. These first in-situ measurements of tropical peat fire emissions should be considered in future models, but the exact implementation scheme is beyond the scope of this paper. Regarding past emissions inventories we

can say the following. Sometimes the values used are a guess that is often not easily recovered from the literature. Koppmann et al. (2005) was never accepted/finished and only discussed gases. Reid et al. (2005) and Andreae and Merlet (2001) do not give peat-specific values. Akagi et al. (2011) give values for peat that are used widely, e.g. in FINN and GFED. Those values were based on one lab fire that we do already compare to extensively (Christian et al., 2003).

Works Cited

Ge, C., Wang, J., and Reid, J. S.: Mesoscale modeling of smoke transport over the Southeast Asian Maritime Continent: coupling of smoke direct radiative effect below and above the low-level clouds, Atmospheric Chemistry and Physics, 14, 159-174, 10.5194/acp-14-159-2014, 2014.

Huijnen, V., Wooster, M., Kaiser, J., Gaveau, D., Flemming, J., Parrington, M., Inness, A., Murdiyarso, D., Main, B., and van Weele, M.: Fire Carbon Emissions Over Maritime Southeast Asia in 2015 Largest Since 1997, Scientific reports, 6, 26886, 2016.

Koplitz, S. N., Mickley, L. J., Marlier, M. E., Buonocore, J. J., Kim, P. S., Liu, T., Sulprizio, M. P., DeFries, R. S., Jacob, D. J., and Schwartz, J.: Public Health Impacts of the Severe Haze in Equatorial Asia in September–October 2015: Demonstration of a New Framework for Informing Fire Management Strategies to Reduce Downwind Smoke Exposure, Environmental Research Letters, 11, 094023, 2016.

Parker, R. J., Boesch, H., Wooster, M. J., Moore, D. P., Webb, A. J., Gaveau, D., and Murdiyarso, D.: Atmospheric CH4 and CO2 Enhancements and Biomass Burning Emission Ratios Derived from Satellite Observations of the 2015 Indonesian Fire Plumes, Atmos. Chem. Phys., 16, 10111-10131, 2016.

Reid, J. S., Hyer, E. J., Johnson, R. S., Holben, B. N., Yokelson, R. J., Zhang, J. L., Campbell, J. R., Christopher, S. A., Di Girolamo, L., Giglio, L., Holz, R. E., Kearney, C., Miettinen, J., Reid, E. A., Turk, F. J., Wang, J., Xian, P., Zhao, G. Y., Balasubramanian,

R., Chew, B. N., Janjai, S., Lagrosas, N., Lestari, P., Lin, N. H., Mahmud, M., Nguyen, A. X., Norris, B., Oanh, N. T. K., Oo, M., Salinas, S. V., Welton, E. J., and Liew, S. C.: Observing and understanding the Southeast Asian aerosol system by remote sensing: An initial review and analysis for the Seven Southeast Asian Studies (7SEAS) program, Atmospheric Research, 122, 403-468, 10.1016/j.atmosres.2012.06.005, 2013.

Stockwell, C. E., Jayarathne, T., Cochrane, M. A., Ryan, K. C., Putra, E. I., Saharjo, B. H., Nurhayati, A. D., Albar, I., Blake, D. R., Simpson, I. J., Stone, E. A., and Yokelson, R. J.: Field measurements of trace gases and aerosols emitted by peat fires in Central Kalimantan, Indonesia, during the 2015 El Nino, Atmospheric Chemistry and Physics, 16, 11711-11732, 10.5194/acp-16-11711-2016, 2016.

Wang, J., Ge, C., Yang, Z. F., Hyer, E. J., Reid, J. S., Chew, B. N., Mahmud, M., Zhang, Y. X., and Zhang, M. G.: Mesoscale modeling of smoke transport over the Southeast Asian Maritime Continent: Interplay of sea breeze, trade wind, typhoon, and topography, Atmospheric Research, 122, 486-503, 10.1016/j.atmosres.2012.05.009, 2013.

---

## Author Comment (AC3) · 15 Dec 2017

Please see "Response to Referee #1" above for the responses and changes to the manuscript in response to this review.
* * *